# LANGUAGE MODEL-DRIVEN DATA PRUNING ENABLES EFFICIENT ACTIVE LEARNING

## ABSTRACT

Active learning (AL) optimizes data labeling efficiency by selecting the most informative instances for annotation. A key component in this procedure is an acquisition function that guides the selection process and identifies the suitable instances for labeling from the unlabeled pool. However, these acquisition methods suffer from high computational costs with large unlabeled data pools, posing a roadblock to their applicability on large datasets. To address this challenge and bridge this gap, we introduce a novel plug-and-play unlabeled data pruning strategy, `ActivePrune`, which leverages language models to prune the unlabeled pool. `ActivePrune` implements a two-stage pruning process: an initial fast evaluation using perplexity scores from an $n$-gram language model, followed by a high-quality selection using metrics for data quality computed through a quantized LLM. Additionally, to enhance the diversity in the unlabeled pool, we propose a novel perplexity reweighting method that systematically brings forward underrepresented instances for selection in subsequent labeling iterations. Experiments on translation, sentiment analysis, topic classification, and summarization tasks on four diverse datasets and four active learning strategies demonstrate that `ActivePrune` outperforms existing data pruning methods. Finally, we compare the selection quality $\leftrightarrow$ efficiency tradeoff of the data pruning methods and demonstrate that `ActivePrune` is computationally more efficient than other LLM score-based pruning methods, and provides up to 74% reduction in the end-to-end time required for active learning.

## 1 INTRODUCTION

Active learning (AL) optimizes data labeling efficiency by selecting the most informative instances of an unlabeled dataset for annotation, thereby enhancing model performance with minimal labeled data (Tsvigun et al., 2022b). This plays a crucial role in reducing labeling efforts and the associated costs by strategically choosing examples for annotation. For example, active learning can help medical experts efficiently prioritize which clinical texts should be labeled first, saving up to 30% annotation time while training a model for clinical named entity recognition (Wei et al., 2019).

The core challenge in active learning is determining which examples from the pool of unlabeled data should be labeled. For this purpose, different acquisition functions, including uncertainty and diversity heuristics, are used to guide the selection process (Zhang et al., 2023). However, as the size of the unlabeled pool increases, the computational cost and latency of the acquisition functions escalate significantly (Maekawa et al., 2022; Tsvigun et al., 2022b; 2023). High acquisition latency leads to several issues: it significantly hinders the *interactivity* of the AL process by increasing the waiting time of experts, leads to higher computational requirements, and thus causes the overall cost of the labeling procedure to increase (Maekawa et al., 2022).

In this work, we thus ask the question, *"Can we efficiently reduce the size of the unlabeled pool without compromising selection quality in active learning?"*. To this end, a few heuristics have been proposed. The simplest and widely used method is random selection (Maekawa et al., 2022), where random instances from the unlabeled pool are selected before applying the acquisition strategy. A better subsampling method, called unlabeled pool subsampling (Tsvigun et al., 2022b), selectively updates uncertainty scores for a subset of instances. In each AL iteration, a portion of the most uncertain instances from preceding iterations is selected, along with a limited number of instances

Figure 1: Illustration of the proposed `ActivePrune` framework for data pruning in Active Learning. Perplexity scores are first computed for the entire unlabeled pool through the KenLM 5-gram model, followed by the computation of data quality scores on a subset of examples through a quantized LLM. Then, the data pruning strategy leverages both these scores to prune the unlabeled pool and send it as the input to the AL acquisition function. After each iteration, a reweighting algorithm adjusts the perplexity distribution based on the selected examples to enhance the diversity for the next iteration.

based on their position in a sorted list by uncertainty. However, this technique cannot be applied reliably in early AL iterations, as the acquisition model is only trained on a small amount of data, which leads to irregular uncertainty estimates (Tsvigun et al., 2022b). Thus, there is still a need for a *generalizable* and *high-quality* data pruning mechanism for reducing the size of the unlabeled pool to enable efficient active learning.

**Our approach.** Recently, large language models (LLMs) have emerged as promising evaluators of data quality (Kocmi & Federmann, 2023; Mekala et al., 2024; Sachdeva et al., 2024). However, LLM quality scores are expensive to compute (Sachdeva et al., 2024) and cannot be directly used for pruning the unlabeled pool in the AL pipeline. To bridge this gap, we propose a novel plug-and-play module called `ActivePrune`, for efficiently pruning the unlabeled pool in AL (Fig. 1). `ActivePrune` operates through a two-stage pruning process, followed by a reweighting procedure. The first stage involves pruning the entire unlabeled pool through a fast evaluation using perplexity scores from an *n*-gram language model (KenLM). In the second stage, a high-quality selection for a smaller set from the remaining pool is done through an open-source quantized LLM (with 2B parameters). The remaining unlabeled pool is then passed to the acquisition function of the AL strategy. To increase diversity in the unlabeled pool for subsequent iterations, we propose a novel perplexity reweighting process to systematically prioritize underrepresented instances for selection and then present a theoretical basis for this method. To validate the effectiveness of `ActivePrune`, we conduct experiments on translation, sentiment analysis, topic classification, and summarization tasks on four diverse datasets and four active learning methods, and demonstrate that our approach outperforms existing unlabeled data selection algorithms across multiple AL strategies. Finally, we compare the selection quality $\leftrightarrow$ efficiency tradeoff of the data pruning methods in AL and demonstrate that `ActivePrune` performs better than other score-based pruning methods (Perplexity, ASK-LLM) while being 97% computationally more efficient compared to such methods. Moroever, `ActivePrune` provides up to 74% reduction in the end-to-end AL time, thus enabling efficient active learning.

**Implications.** Our work provides a high-quality AL-strategy agnostic method of reducing the size of large datasets so that they can be used in active learning. This enables more efficient utilization of computational resources and increases the *interactivity* of the labeling process, thereby saving the time of labeling experts. Finally, it paves the way for broader applicability of active learning techniques to high-volume, domain-specific datasets.

## 2 RELATED WORK

Our work brings together two closely related but distinct (and well-studied) domains of data pruning and active learning. Thus, we provide an overview of the literature on these two areas and then briefly describe the existing approaches of large language models for data quality assessment and pruning.

### 2.1 ACTIVE LEARNING

Active learning (AL) techniques focus on selectively querying unlabeled data that, when labeled, are most beneficial for model training (Ren et al., 2021; Deng et al., 2023; Yuan et al., 2020). This

selective data querying aims to reduce annotation costs while improving model performance. Pool-based active learning methods are primarily uncertainty-based (Zhao et al., 2020), diversity-based (Hu & Neubig, 2021; Pendas et al., 2023; Zhang et al., 2022), or hybrid (Azeemi et al., 2024; Margatina et al., 2021a; Dor et al., 2020; Korakakis et al., 2024). Despite the improvements in query strategies, the trade-off between high-quality acquisition and computational overhead remains a central issue in AL research. Approaches like Bayesian query strategies offer sophisticated means to quantify uncertainty but at a significant computational and memory cost (Siddhant & Lipton, 2018). This has led to a continued search for methods that balance informative sample selection with practical constraints on the computation and reduce the waiting time between iterations.

## 2.2 Data Pruning

Data Pruning focuses on the elimination of redundant or less informative samples from the training dataset to reduce computational costs in training (Kothawade et al., 2021; Killamsetty et al., 2021b; Kothyari et al., 2021; Azeemi et al., 2022; Schreiber et al., 2020; Lee et al., 2021; Sorscher et al., 2022), improve model generalization (Paul et al., 2021; Marion et al., 2023; Azeemi et al., 2023a; Sachdeva et al., 2024), or reduce bias and remove harmful content. Different metrics are used in data pruning techniques to assess the informativeness or redundancy of data samples, such as geometry-based (Agarwal et al., 2020; Sener & Savarese, 2018), uncertainty-based (Coleman et al., 2019), margin-based (Park et al., 2022), gradient-matching metrics (Mirzasoleiman et al., 2020; Killamsetty et al., 2021a), forgetting scores (Toneva et al., 2018; Azeemi et al., 2022), and metrics based on training trajectories (Paul et al., 2021). Despite their utility, these methods often face challenges regarding their generalizability across different architectures or datasets and may incur non-negligible overheads (Paul et al., 2021; Bartoldson et al., 2023). This can be mitigated through self-supervised pruning metrics, which do not require initial training runs (Sorscher et al., 2022; Azeemi et al., 2023b), or dynamic data pruning methods, which select samples throughout training based on their relevance (Raju et al., 2021; He et al., 2023; Qin et al., 2023). In contrast to the existing methods, our focus in this work is to develop a novel data pruning mechanism that is specifically tailored for active learning.

## 2.3 LLMs for Data Filtering

Recent works have leveraged large language models (LLMs) for data quality assessment, opening up new dimensions for optimizing training datasets. Techniques focusing on quality-score sampling, as discussed by Maini et al. (2024) and Engstrom et al. (2024), employ LLMs to evaluate and select high-quality training samples based on various criteria, including perplexity filtering (Marion et al., 2023; Muennighoff et al., 2024; Ankner et al., 2024) and similarity to high-quality data sources (Xie et al., 2023; Anil et al., 2023; Javaheripi et al., 2023). These methods aim to prioritize samples that are representative or have desirable characteristics, thereby enhancing the overall quality of the training dataset. Furthermore, model-based training-run simulators (Guu et al., 2023) and submodular selection routines (Coleman et al., 2019) represent newer approaches to assess and select the training data to increase the efficiency of downstream fine-tuning.

## 3 Method

In this section, we introduce `ActivePrune`, a framework for efficient pruning of the unlabeled pool to enable computationally feasible active learning. `ActivePrune` has the following key components:

1. A fast, approximate quality evaluation of the entire unlabeled pool using perplexity scores (Section 3.2).

2. A high-quality assessment for a select subset of candidates through a quantized LLM (Section 3.3).

3. A reweighting algorithm that incorporates the labeled instances after each AL iteration to reweight the perplexity distribution and promote the selection of examples from previously underrepresented subpopulations (Section 3.4).

## 3.1 PRELIMINARIES

We follow the standard active learning setup used in previous works (Hu & Neubig, 2021; Tsvigun et al., 2022b). Consider an unlabeled dataset $\mathcal{U} = \{x_i\}_{i=1}^{N}$ and the labeled set $\mathcal{L}$ which is initially empty. In each iteration, the acquisition strategy $\mathcal{Q}$ selects the *most informative* examples for annotation which are then annotated by human experts $\mathcal{H}(\cdot)$. During the experiments, we simulate this labeling procedure by acquiring the actual labels from the dataset, following the protocol from previous works (Tsvigun et al., 2022b; Hu & Neubig, 2021; Tsvigun et al., 2023). The labeled examples are used to train the acquisition model $\mathcal{M}$, added to the labeled dataset $\mathcal{L}$, and excluded from the unlabeled dataset $\mathcal{U}$. The model is then evaluated on an unseen validation set. This process continues until the budget $b$ is exhausted. In each iteration, we construct a filtered unlabeled pool $\mathcal{F}$ using `ActivePrune`, which is used by the acquisition strategy instead of the complete unlabeled pool $\mathcal{U}$.

## 3.2 PERPLEXITY CALCULATION

The first step in `ActivePrune` is to perform a quality evaluation for the complete unlabeled pool $\mathcal{U}$. For this purpose, we first compute the perplexity scores for each example in the unlabeled pool $\mathbf{P} = \{p_1, p_2, \ldots, p_n\}$. This process must be computationally feasible so that large datasets are processed quickly. Thus, for fast and efficient perplexity calculation, we use the KenLM 5-gram language model (Heafield, 2011) along with the SentencePiece tokenizer (Kudo & Richardson, 2018). KenLM enables fast and memory-efficient language model queries through the `Probing` and `Trie` data structures, and uses a modified version of Kneser-Ney smoothing. The `Probing` variant of KenLM achieves a computation speed of 1818 perplexity queries per millisecond (Heafield, 2011) on the English Gigaword corpus (Parker, Robert et al., 2009). We use the KenLM trained on the Oscar dataset (Suárez et al., 2020) derived from CommonCrawl, which contains web-crawl data. Using KenLM trained on web-crawl data enables the perplexity filtering procedure to generalize more effectively across diverse domains, enhancing its robustness to various types of datasets, as demonstrated by the results in Table 1.

Once the perplexity scores are computed, we sort them in an ascending order $\mathbf{P}_{\text{sorted}} = \text{argsort}(\mathbf{P}, \text{order} = \text{ascending})$ and select the $1...k$ indices that have the lowest perplexity scores: $\mathcal{P}_s = \{p_i \in \mathbf{P}_{\text{sorted}} \mid i = 1, 2, \ldots, k\}$. The examples with the lowest perplexity scores are then added to the filtered pool: $\mathcal{F} = \mathcal{F} \cup \mathcal{P}_s$. The examples with high perplexity scores are filtered out (based on a pruning percentage $p$) as they represent highly surprising and potentially noisy examples (Muennighoff et al., 2024; Marion et al., 2023; Sachdeva et al., 2024) and these examples are thus re-evaluated through data quality scores in the next step before they can be selected for labeling.

## 3.3 LLM DATA QUALITY SCORES

We compute the data quality scores for the top $m$ high-perplexity examples from the remaining unlabeled pool $x \in \mathcal{U} \setminus \mathcal{P}_s$. We adapt the procedure presented in (Sachdeva et al., 2024) for computing data quality scores through an open-source LLM pre-trained on data from diverse domains. Given the task type $\tau$, the data quality score $q(x_i)$ for each example $x_i$ is computed as follows:

**(1)** — Construct a prompt $P(x_i, \tau)$ with example $x_i$ and the task type $\tau$. The prompt specifically asks whether $x_i$ should be included in the training set for task $\tau$ (Fig. 2) and asks for a response as *"yes"* or *"no"*. **(2)** — Obtain the response $R$ from the LLM, which includes a decision on whether $x_i$ is suitable for inclusion in the training set. This decision is represented by tokens such as *"yes"* or *"no"*. **(3)** — Compute the data quality score $q(x_i)$ for each example $x_i$ as the softmax probability of the token *"yes"* in the LLM's response, $q(x_i)$, computed as:

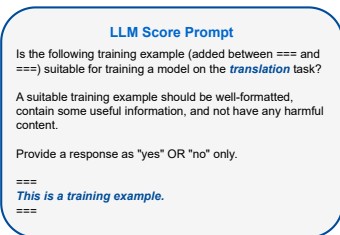

**LLM Score Prompt**

Is the following training example (added between === and ===) suitable for training a model on the *translation* task?

A suitable training example should be well-formatted, contain some useful information, and not have any harmful content.

Provide a response as "yes" OR "no" only.

===
*This is a training example.*
===

$$\text{softmax}(\text{LLM}(P(x_i, \tau)))_{\text{"yes"}} = \frac{e^{\text{LLM}(P(x_i, \tau))_{\text{"yes"}}}}{\sum_j e^{\text{LLM}(P(x_i, \tau))_j}}$$

Figure 2: An example prompt for determining the data quality score for a training example on translation task.

where $\text{LLM}(P(x_i, \tau))$ denotes the vector of logits output by the LLM for the prompt $P(x_i, \tau)$, and $\text{softmax}(\cdot)_{\text{"yes"}}$ indicates the softmax probability corresponding to the token "yes".

Calculating data quality scores through an LLM over a sizable dataset can be computationally expensive. Therefore, we use a quantized LLM for the computation of data quality scores. Quantization (Dettmers et al., 2022) significantly reduces the cost of inference by representing weights in limited bits. We use 4-bit quantized LLMs (Dettmers et al., 2024) which have been shown to match the full 16-bit fine-tuning performance despite using a fraction of memory.

After calculating the data quality scores, we sort them in a descending order $\mathbf{Q}_{\text{sorted}} = \text{argsort}(\mathbf{Q}, \text{order} = \text{descending})$ and select the $1...k$ indices having the highest data quality scores: $\mathcal{Q}_s = \{q_i \in \mathbf{Q}_{\text{sorted}} \mid i = 1, 2, \ldots, k\}$. These are then added to the filtered pool as well: $\mathcal{F} = \mathcal{F} \cup \mathcal{Q}_s$. The final unlabeled pool containing examples selected through perplexity and data quality scores is then passed to the acquisition function which begins the next AL iteration.

### 3.4 Perplexity Reweighting

After each AL iteration, we dynamically adjust the perplexity scores of the unlabeled pool to prioritize diversity in the examples. This adjustment is based on a similarity measure relative to the perplexity scores of the instances most recently added to the labeled set $\mathcal{L}$. The adjustment factor for an instance $x_i$, given the set of newly labeled instances $L \subseteq \mathcal{L}$, is defined as:

$$A(x_i, L) = \frac{1}{|L|} \sum_{l \in L} |P(x_i) - P(l)| \tag{1}$$

This factor measures the average absolute difference in perplexity between $x_i$ and the instances in $L$, reflecting the similarity of $x_i$ to the instances most recently labeled. We use the adjustment factor to compute new perplexity:

$$P_{\text{new}}(x_i) = P_{old}(x_i) - \beta.A(x_i, L) = P_{old}(x_i) - \beta.\frac{1}{|L|} \sum_{l \in L} |P(x_i) - P(l)| \tag{2}$$

The perplexity of the instances in the unlabeled pool which are farther away from the newly labeled instances is decreased after reweighting, thereby increasing their chance of being selected through bottom-$k$ perplexity sampling. $\beta$ is a hyperparameter that controls the sensitivity to the adjustment factor. Higher values of $\beta$ result in a steeper decrease in the perplexity for instances farther away from those recently labeled. Note that perplexity reweighting takes place outside the AL procedure itself and instead within the data pruning step, making it independent of any specific AL strategy. Thus, perplexity reweighting is different from other sample reweighting techniques in AL that are typically designed to increase batch diversity for specific acquisition functions, e.g., BADGE (Ash et al., 2019).

The following proposition outlines the mechanism by which perplexity reweighting enhances diversity across multiple iterations.

**Proposition 1.** *Let $\mathcal{U} = \{x_i\}_{i=1}^N$ be the unlabeled pool and $\mathcal{L}_t$ be the labeled set at iteration $t$. Let $S \subseteq \mathcal{U}$ represent an underrepresented subset, containing the instances in $S$ that have perplexity scores significantly different from those in $\mathcal{L}_t$. Under the perplexity reweighting procedure, for any $x_i \in S$, the probability of selecting $x_i$ increases progressively, ensuring its selection within a finite number of iterations.*

The detailed proof for this proposition is included in Appendix A, which shows that the perplexity reweighting mechanism prevents the data pruning process from focusing narrowly, as underrepresented instances are systematically brought forward for selection. This is done progressively by reducing the perplexity of such instances over multiple AL iterations, which in turn guarantees that these are selected with high probability within a finite number of AL iterations, thereby promoting diversity in the selected data. In practical scenarios, although the total number of AL iterations may be constrained by the available budget, perplexity-based reweighting still ensures that underrepresented instances are consistently prioritized for selection.

# 4 EXPERIMENTAL SETUP

## 4.1 DATASETS

We evaluate our method on four diverse datasets spanning different tasks and domains, including sentiment analysis, topic classification, translation and abstractive text summarization:

**IMDB** dataset (Maas et al., 2011) for binary classification task with movie reviews. It is utilized for assessing sentiment analysis models, consisting of 25,000 reviews for training and an equal number for testing.

**AG News** dataset (Zhang et al., 2015) for topic classification task with 4 classes. The dataset includes 120,000 training samples and 7,600 test samples across the categories of World, Sports, Business, and Science/Technology.

**IT** domain dataset (Koehn & Knowles, 2017) for En-De machine translation. We use the deduplicated and resplit version (Aharoni & Goldberg, 2020) containing 223,000 sentence pairs for training and 2,000 pairs for testing.

**AESLC** dataset (Zhang & Tetreault, 2019) consisting of email threads for the task of abstractive text summarization. Each instance contains an email and a subject line as the corresponding summary. The training set comprises approximately 14,400 email threads, while the validation and test sets contain around 1960 and 1910 email threads, respectively.

## 4.2 MODELS

For the abstractive text summarization and machine translation tasks, we utilize the **BART** model in our experiments (Lewis et al., 2019) due to its strong performance on these tasks. BART constitutes a transformer encoder-decoder architecture with a bidirectional encoder and an autoregressive decoder. We use the BART-base variant with 139M parameters.

For the topic classification and sentiment analysis tasks, we use the **RoBERTa** model (Liu et al., 2019), an encoder-only transformer language model. We use the RoBERTa-base variant containing 125M parameters.

For computing the LLM data quality scores and the LLM perplexity scores, we use the **Gemma-2B** LLM (Team et al., 2024), an open-source LLM based on the Gemini research (Team et al., 2023). Gemma-2B is a decoder-only transformer language model with 18 layers and 2B parameters.

## 4.3 ACTIVE LEARNING METHODS

We consider several active learning methods for evaluation.

**Random Sampling** draws training examples uniformly at random and is considered a competitive baseline in AL (Hu & Neubig, 2021; Tsvigun et al., 2023).

**Least Confidence** (Lewis, 1995) selects the examples on which the model is least confident.

**Coreset** (Sener & Savarese, 2017) samples based on representations extracted from the model's penultimate layer.

**Normalized Sequence Probability (NSP)** (Ueffing & Ney, 2007) is an uncertainty-based strategy designed to measure the uncertainty in a model's predictions on a specific sequence. It forms the basis for numerous other uncertainty-based active learning (AL) methods.

**In-Domain Diversity Sampling (IDDS)** (Tsvigun et al., 2023) aims to choose sentences distinct from those previously annotated to enhance the diversity of the final collection. At the same time, it avoids picking noisier instances that differ significantly from the documents in the in-domain dataset.

## 4.4 BASELINES

We compare our method with different unlabeled pool subsampling and data pruning strategies. **(1) Random Selection** selects examples randomly from the unlabeled pool which are then passed

Table 1: Comparison of unlabeled pool pruning methods for Active Learning. For each dataset, we run experiments on two active learning methods across five pool pruning strategies and evaluate the performance on the validation set. Five active learning iterations are done and 1% of the dataset is selected for labeling in each iteration, based on the protocol from prior works. For each strategy, we do three runs with different seeds and report the mean evaluation score, with the standard error of the mean given in the subscript. The **bold** values represent top-performing results compared to other data pruning strategies on a specific task and AL method.

| DATASET (TASK) | EVALUATION METRIC | AL METHOD | DATA PRUNING STRATEGY | AL ITERATION | | | | |
|---|---|---|---|---|---|---|---|---|
| | | | | 1 | 2 | 3 | 4 | 5 |
| IT DOMAIN (Translation) | SacreBLEU | IDDS | No Pruning | $25.43_{\pm0.0}$ | $25.22_{\pm0.0}$ | $24.98_{\pm0.0}$ | $25.12_{\pm0.0}$ | $25.20_{\pm0.0}$ |
| | | | Random | $25.38_{\pm0.1}$ | $24.95_{\pm0.1}$ | $24.82_{\pm0.1}$ | $24.96_{\pm0.1}$ | $25.00_{\pm0.1}$ |
| | | | ASK-LLM | $25.25_{\pm0.1}$ | $25.08_{\pm0.1}$ | $24.87_{\pm0.1}$ | $24.96_{\pm0.1}$ | $25.06_{\pm0.1}$ |
| | | | Perplexity | $25.63_{\pm0.1}$ | $24.81_{\pm0.1}$ | $24.78_{\pm0.1}$ | $24.79_{\pm0.1}$ | $24.72_{\pm0.1}$ |
| | | | UPS | $25.21_{\pm0.1}$ | $24.42_{\pm0.1}$ | $24.32_{\pm0.1}$ | $24.41_{\pm0.1}$ | $24.22_{\pm0.1}$ |
| | | | ActivePrune | $\mathbf{27.01}_{\pm0.1}$ | $\mathbf{27.39}_{\pm0.1}$ | $\mathbf{27.80}_{\pm0.1}$ | $\mathbf{28.40}_{\pm0.1}$ | $\mathbf{28.97}_{\pm0.1}$ |
| | | NSP | No Pruning | $27.89_{\pm0.0}$ | $28.61_{\pm0.0}$ | $29.42_{\pm0.0}$ | $30.10_{\pm0.0}$ | $30.57_{\pm0.0}$ |
| | | | Random | $27.94_{\pm0.0}$ | $28.95_{\pm0.1}$ | $29.51_{\pm0.4}$ | $29.87_{\pm0.2}$ | $30.09_{\pm0.2}$ |
| | | | ASK-LLM | $28.07_{\pm0.1}$ | $29.39_{\pm0.1}$ | $29.59_{\pm0.2}$ | $30.13_{\pm0.1}$ | $30.46_{\pm0.1}$ |
| | | | Perplexity | $27.86_{\pm0.1}$ | $29.07_{\pm0.1}$ | $29.53_{\pm0.3}$ | $30.15_{\pm0.2}$ | $30.53_{\pm0.2}$ |
| | | | UPS | $27.82_{\pm0.1}$ | $29.02_{\pm0.2}$ | $29.69_{\pm0.0}$ | $30.33_{\pm0.1}$ | $30.49_{\pm0.1}$ |
| | | | ActivePrune | $\mathbf{28.70}_{\pm0.0}$ | $\mathbf{29.85}_{\pm0.2}$ | $\mathbf{30.52}_{\pm0.1}$ | $\mathbf{30.76}_{\pm0.2}$ | $\mathbf{30.97}_{\pm0.1}$ |
| AESLC (Summarization) | ROUGE-L | IDDS | No Pruning | $29.14_{\pm0.0}$ | $29.66_{\pm0.0}$ | $28.91_{\pm0.0}$ | $30.10_{\pm0.0}$ | $29.30_{\pm0.0}$ |
| | | | Random | $28.03_{\pm0.4}$ | $29.02_{\pm0.1}$ | $28.31_{\pm0.2}$ | $29.25_{\pm0.2}$ | $29.96_{\pm0.1}$ |
| | | | ASK-LLM | $28.15_{\pm0.4}$ | $28.99_{\pm0.3}$ | $28.86_{\pm0.2}$ | $29.54_{\pm0.5}$ | $29.00_{\pm0.1}$ |
| | | | Perplexity | $28.54_{\pm0.4}$ | $29.13_{\pm0.3}$ | $29.68_{\pm0.2}$ | $29.15_{\pm0.3}$ | $29.93_{\pm0.3}$ |
| | | | UPS | $28.34_{\pm0.4}$ | $28.71_{\pm0.4}$ | $29.46_{\pm0.5}$ | $29.13_{\pm0.0}$ | $30.14_{\pm0.4}$ |
| | | | ActivePrune | $\mathbf{28.76}_{\pm0.2}$ | $\mathbf{30.08}_{\pm0.2}$ | $\mathbf{29.94}_{\pm0.1}$ | $\mathbf{30.41}_{\pm0.2}$ | $\mathbf{30.85}_{\pm0.2}$ |
| | | NSP | No Pruning | $28.78_{\pm0.0}$ | $29.10_{\pm0.0}$ | $29.79_{\pm0.0}$ | $29.97_{\pm0.0}$ | $30.47_{\pm0.0}$ |
| | | | Random | $28.03_{\pm0.4}$ | $28.69_{\pm0.3}$ | $28.62_{\pm0.4}$ | $29.81_{\pm0.3}$ | $29.53_{\pm0.2}$ |
| | | | ASK-LLM | $27.62_{\pm0.4}$ | $28.72_{\pm1.0}$ | $29.26_{\pm0.6}$ | $29.25_{\pm0.6}$ | $30.07_{\pm0.3}$ |
| | | | Perplexity | $27.96_{\pm0.4}$ | $28.44_{\pm0.2}$ | $29.02_{\pm0.4}$ | $29.46_{\pm0.2}$ | $29.53_{\pm0.1}$ |
| | | | UPS | $28.02_{\pm0.4}$ | $29.04_{\pm0.1}$ | $29.69_{\pm0.4}$ | $29.41_{\pm0.6}$ | $29.80_{\pm0.1}$ |
| | | | ActivePrune | $\mathbf{28.04}_{\pm0.1}$ | $\mathbf{29.46}_{\pm0.1}$ | $\mathbf{29.89}_{\pm0.4}$ | $\mathbf{30.29}_{\pm0.2}$ | $\mathbf{30.34}_{\pm0.2}$ |
| IMDB (Classification) | F1-Score | Coreset | No Pruning | $91.90_{\pm0.0}$ | $92.36_{\pm0.0}$ | $92.32_{\pm0.0}$ | $92.68_{\pm0.0}$ | $93.01_{\pm0.0}$ |
| | | | Random | $91.35_{\pm0.1}$ | $92.11_{\pm0.3}$ | $92.40_{\pm0.1}$ | $92.83_{\pm0.1}$ | $92.99_{\pm0.1}$ |
| | | | ASK-LLM | $91.25_{\pm0.5}$ | $91.81_{\pm0.7}$ | $92.00_{\pm1.0}$ | $92.09_{\pm0.9}$ | $91.64_{\pm0.2}$ |
| | | | Perplexity | $91.41_{\pm0.1}$ | $92.16_{\pm0.0}$ | $92.31_{\pm0.1}$ | $92.66_{\pm0.1}$ | $92.91_{\pm0.0}$ |
| | | | UPS | $87.28_{\pm0.7}$ | $91.89_{\pm0.2}$ | $92.40_{\pm0.0}$ | $92.83_{\pm0.1}$ | $93.03_{\pm0.1}$ |
| | | | ActivePrune | $\mathbf{91.67}_{\pm0.1}$ | $\mathbf{92.20}_{\pm0.1}$ | $\mathbf{92.60}_{\pm0.4}$ | $\mathbf{92.84}_{\pm0.1}$ | $\mathbf{93.15}_{\pm0.1}$ |
| | | LC | No Pruning | $91.68_{\pm0.0}$ | $92.10_{\pm0.0}$ | $92.92_{\pm0.0}$ | $92.46_{\pm0.0}$ | $92.98_{\pm0.0}$ |
| | | | Random | $92.03_{\pm0.1}$ | $92.07_{\pm0.1}$ | $92.19_{\pm0.1}$ | $92.68_{\pm0.0}$ | $92.53_{\pm0.1}$ |
| | | | ASK-LLM | $90.13_{\pm0.2}$ | $90.26_{\pm0.9}$ | $88.30_{\pm0.4}$ | $90.15_{\pm0.2}$ | $90.33_{\pm0.1}$ |
| | | | Perplexity | $91.71_{\pm0.1}$ | $92.08_{\pm0.1}$ | $92.18_{\pm0.2}$ | $92.74_{\pm0.1}$ | $92.79_{\pm0.0}$ |
| | | | UPS | $89.87_{\pm0.1}$ | $91.71_{\pm0.1}$ | $92.29_{\pm0.2}$ | $92.71_{\pm0.1}$ | $92.79_{\pm0.0}$ |
| | | | ActivePrune | $\mathbf{92.37}_{\pm0.0}$ | $\mathbf{92.46}_{\pm0.1}$ | $\mathbf{92.76}_{\pm0.0}$ | $\mathbf{93.16}_{\pm0.1}$ | $\mathbf{93.11}_{\pm0.1}$ |
| AG NEWS (Classification) | F1-Score | Coreset | No Pruning | $92.36_{\pm0.0}$ | $92.91_{\pm0.0}$ | $93.50_{\pm0.0}$ | $93.80_{\pm0.0}$ | $94.05_{\pm0.0}$ |
| | | | Random | $92.04_{\pm0.2}$ | $93.30_{\pm0.1}$ | $94.00_{\pm0.1}$ | $94.10_{\pm0.1}$ | $94.06_{\pm0.1}$ |
| | | | ASK-LLM | $91.71_{\pm0.2}$ | $92.45_{\pm0.2}$ | $93.18_{\pm0.2}$ | $93.65_{\pm0.2}$ | $93.83_{\pm0.1}$ |
| | | | Perplexity | $\mathbf{92.64}_{\pm0.2}$ | $93.24_{\pm0.2}$ | $93.41_{\pm0.1}$ | $93.59_{\pm0.2}$ | $93.93_{\pm0.1}$ |
| | | | UPS | $92.29_{\pm0.2}$ | $92.96_{\pm0.1}$ | $93.70_{\pm0.1}$ | $94.07_{\pm0.1}$ | $94.04_{\pm0.1}$ |
| | | | ActivePrune | $92.40_{\pm0.2}$ | $\mathbf{93.55}_{\pm0.1}$ | $\mathbf{94.13}_{\pm0.1}$ | $\mathbf{94.33}_{\pm0.1}$ | $\mathbf{94.22}_{\pm0.1}$ |
| | | LC | No Pruning | $92.58_{\pm0.0}$ | $93.40_{\pm0.0}$ | $93.98_{\pm0.0}$ | $93.98_{\pm0.0}$ | $94.40_{\pm0.0}$ |
| | | | Random | $91.85_{\pm0.1}$ | $92.75_{\pm0.1}$ | $93.53_{\pm0.1}$ | $93.91_{\pm0.1}$ | $94.08_{\pm0.1}$ |
| | | | ASK-LLM | $91.20_{\pm0.1}$ | $92.43_{\pm0.1}$ | $93.33_{\pm0.1}$ | $93.72_{\pm0.1}$ | $93.86_{\pm0.1}$ |
| | | | Perplexity | $\mathbf{92.54}_{\pm0.1}$ | $\mathbf{93.15}_{\pm0.1}$ | $93.45_{\pm0.1}$ | $93.56_{\pm0.1}$ | $93.85_{\pm0.1}$ |
| | | | UPS | $92.46_{\pm0.1}$ | $93.07_{\pm0.1}$ | $93.34_{\pm0.1}$ | $93.85_{\pm0.1}$ | $93.92_{\pm0.1}$ |
| | | | ActivePrune | $91.69_{\pm0.1}$ | $92.98_{\pm0.1}$ | $\mathbf{93.89}_{\pm0.1}$ | $\mathbf{93.93}_{\pm0.1}$ | $\mathbf{94.30}_{\pm0.1}$ |

to the active learning method. **(2) ASK-LLM** selects the top-k examples with the highest quality scores assigned by an LLM (Sachdeva et al., 2024). **(3) Perplexity** picks examples with the lowest perplexity scores (computed through an LLM) in an attempt to exclude noisy examples and label noise (Laurençon et al., 2022). **(4) UPS (Unlabeled pool subsampling)** prefers the instances with the highest uncertainty scores from the unlabeled pool and selectively updates those in each AL iteration (Tsvigun et al., 2022b).

### 4.5 TRAINING SETTINGS

For the translation task on the IT domain dataset, we use the BART-base variant (with 139 million parameters) trained on the WMT14 En-De out-of-domain parallel corpora, following the setup in previous works (Dou et al., 2019; Hu & Neubig, 2021). SacreBLEU score is used for the evaluation on the test set of the IT domain dataset. For abstractive text summarization on the AESLC dataset, we follow the setup used in Tsvigun et al. (2023) and train the BART-base model. ROUGE-L score is used for the evaluation on the test set of AESLC. For binary sentiment classification on the IMDB dataset and topic classification on the AG News dataset, we follow the protocol from Tsvigun et al. (2022b) and fine-tune the RoBERTa-base model. F1-score is used for the evaluation of the model on the test set of IMDB and AG News datasets.

In each AL iteration, the acquisition model is trained for 5 epochs, with a learning rate of 2e-5, weight decay of 0.028, and a batch size of 16. Gradient clipping was set at 0.28, and a warmup steps factor of 0.1 was applied to the learning rate scheduler. For the 4-bit quantization of the Gemma-2B model, we use the CUDA implementation provided by `bitsandbytes` [1].

At the start of each iteration, the data pruning strategy selects 25% of the dataset (thus pruning away 75% of the dataset), which is then passed to the AL strategy. 1% of the dataset is selected by the AL strategy, added to the labeled pool (i.e., emulating the labeling process), and then used for training the acquisition model, mirroring the protocol presented in Tsvigun et al. (2022b). $\beta$ (perplexity reweighting sensitivity) is set to 0.1 after tuning through validation set. We evaluate the acquisition model on the unseen test dataset after each AL iteration.

## 5 EXPERIMENTS

### 5.1 RESULTS

We compare `ActivePrune` with the other four data pruning strategies in Table 1. For each combination of the dataset, AL strategy, and data pruning method, we do three runs with different seeds and report the mean evaluation metric along with the standard error. Across all datasets, `ActivePrune` performs better than other strategies in the final AL iteration with *statistically significant* differences. On the IT domain dataset with IDDS strategy, `ActivePrune` shows the highest SacreBLEU consistently over 5 AL iterations, with an increase of 15.61% (25.06 to 28.97) after the final iteration compared to the runner-up baseline (ASK-LLM). For the NSP strategy, `ActivePrune` shows an increase of 1.44% SacreBLEU over Perplexity (30.53 to 30.97) to 2.92% over Random (30.09 to 30.97). For the AESLC dataset, the difference in ROUGE-L score between `ActivePrune` and other strategies goes up to 6.37%. In certain configurations, the performance of `ActivePrune` even exceeds the *no pruning* baseline. For instance, our method achieves a SacreBLEU score of 30.97 on the IT domain dataset with NSP strategy, compared to 30.57 SacreBLEU without pruning. We hypothesize that this performance increase results from the selective removal of noisy examples from the entire unlabeled pool.

For the AG News and IMDB classification tasks, `ActivePrune` demonstrates the highest F1-score across most AL iterations for both Coreset and Least Confidence strategies. Note that the relative performance of other strategies varies between datasets, pointing towards differences in the data distribution. For example, ASK-LLM does not perform better than Perplexity and UPS on IMDB dataset. However, `ActivePrune` is able to perform consistently well, indicating the robustness of this strategy to different domains and distribution shifts.

### 5.2 COMPUTATIONAL COST COMPARISON

**Comparison with other pruning methods.** We now compare the computational efficiency of `ActivePrune` with other unlabeled data pruning methods (Random, UPS, ASK-LLM, Perplexity) under the same hardware and software configuration (Table 2). We conduct this analysis on the IT domain dataset, as it is the largest dataset (with 223,000 training instances) on which we conducted our experiments. We focus on the time taken (s) for unlabeled data selection, the acquisition and training time, and the overall end-to-end time for the complete AL procedure for 5 iterations. `ActivePrune` is

---

[1]https://github.com/TimDettmers/bitsandbytes

97.5% more efficient in pruning the unlabeled pool compared to Perplexity and by 97.6% compared to ASK-LLM. Specifically, the pruning time decreased from 34,550s for Perplexity and 36,010s for ASK-LLM to just 840s for `ActivePrune`. Note that the high computational cost of ASK-LLM and Perplexity pruning baseline is primarily due to the expensive LLM inference on the *entire unlabeled pool*, in contrast to `ActivePrune` which adopts a two-stage pruning process: the usage of KenLM $n$-gram language model in the first stage and the quantized LLM inference for a limited number of examples in the second stage. KenLM shows a significantly lower computational cost for perplexity calculation, with 1818 perplexity queries per millisecond, which is not possible with a regular LLM.

The increase in efficiency is mirrored in the total time as well, with `ActivePrune` showing a decrease of 44.1% and 45.2% in the active learning time compared to Perplexity and ASK-LLM, respectively. Despite being computationally more efficient, `ActivePrune` demonstrates a higher SacreBLEU score on the test set.

**Enabling efficient active learning.** The end-to-end AL procedure without data pruning on the NSP strategy and the complete IT domain dataset requires 166,275s. In contrast, ActivePrune takes 42,240s (Table 2), which is a 74.5% decrease in the time required for Active Learning. This highlights the benefits of incorporating data pruning in the AL procedure and demonstrates that it enables efficient active learning.

Table 2: Performance and wall clock time (seconds) comparison of the end-to-end active learning procedure for five iterations across different unlabeled pool pruning methods. The NSP strategy was used on the IT domain dataset on a single A6000 GPU server.

|  | Random | UPS | Perplexity | ASK-LLM | ActivePrune (Ours) |
|---|---|---|---|---|---|
| SacreBLEU | 30.09 | 30.49 | 30.53 | 30.46 | 30.97 |
| Unlabeled pool pruning (s) | 12 | 20 | 34,550 | 36,010 | 840 |
| Acquisition and training time (s) | 41,400 | 41,400 | 41,400 | 41,400 | 41,400 |
| Total time (s) | 41,412 | 41,420 | 75,950 | 77,410 | 42,240 |

**Selection quality ↔ efficiency tradeoff.** We present the trade-off between selection quality and efficiency for AL data pruning strategies in Fig. 3. The shaded regions indicate efficiency categories with light gray representing cost-effective methods, light blue for high-quality methods that require more time, and light green marking preferred region. Our method emerges as a notably efficient strategy, with a SacreBLEU score of 30.97 and requiring significantly less time than more expensive metrics. Thus, `ActivePrune` optimizes the Pareto frontier of the example selection quality and the required computational resources for pruning the unlabeled pool in AL.

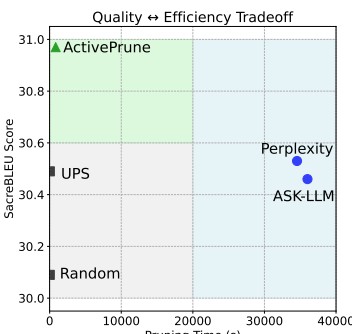

Figure 3: Trade-off between quality and efficiency across various data pruning strategies on the IT domain dataset with the NSP acquisition method. The perplexity method in the figure reflects the perplexity scores computed through the LLM (Gemma-2B) and not the perplexity scores from KenLM (which is a component of `ActivePrune` only).

## 5.3 EXAMINING THE SELECTION

To understand the sentence selection preference in `ActivePrune` during AL iterations, it is important to find out how the quality of the selected instances differs from other data pruning methods. We plot LLM data quality vs. perplexity scores and analyze the unlabeled instances selected in ten iterations for annotation by each unlabeled pool pruning strategy (Random, UPS, `ActivePrune`, Perplexity, ASK-LLM) on the IMDB dataset. Figure 4 shows that Random and UPS strategies do not prioritize instances having a high data quality score or a high perplexity score (i.e., orange instances). A few examples of such instances are presented in Appendix D. Conversely, `ActivePrune`, Perplexity, and ASK-LLM select the instances having high scores, in addition to prototypical examples (blue examples). `ActivePrune` has the additional benefit of being significantly more efficient than the other score-based strategies (Section 5.2).

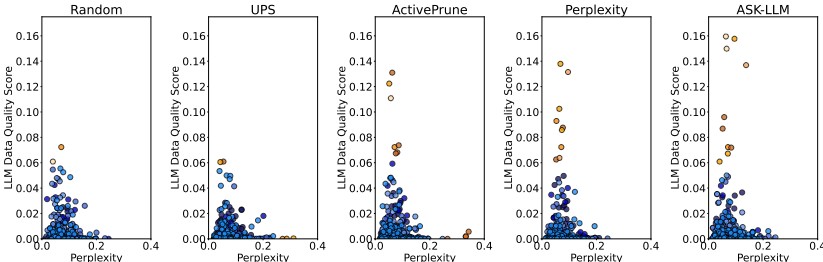

Figure 4: Distribution of Perplexity vs LLM data quality scores across samples selected through different pruning methods for the IMDB dataset. Each subplot represents sentences selected through the Active Learning (AL) procedure across 10 iterations, with each iteration involving the selection 1% of the dataset. Examples with orange color indicate examples with a *high* LLM quality score or a *high* perplexity score. The color gradient indicates the sequence of iterations, where the darkest shade denotes the first iteration and progressively lighter shades denote subsequent iterations, up to the lightest shade for the tenth iteration. Subplots are organized by sampling strategy.

## 5.4 ABLATION EXPERIMENTS

We conduct ablation experiments to demonstrate the importance of different components and algorithmic choices in `ActivePrune`. These experiments are conducted on the AESLC summarization dataset with the IDDS AL strategy.

**Analyzing different components.** We analyze the impact of three key components in `ActivePrune`: Perplexity, LLM Score, and Reweighting (Table 3). The configuration which includes only Perplexity and Reweighting achieves a ROUGE-L score of 30.05. However, this combination alone does not fully exploit the potential of `ActivePrune`. Next, we assess the effect of incorporating the LLM Score without reweighting, resulting in a ROUGE-L score of 30.34. This significant improvement suggests that the LLM Score plays a crucial role in enhancing the model's performance.

Further, excluding Perplexity while retaining the LLM Score yields a ROUGE-L score of 29.00. This configuration does not perform better than scenario where both the Perplexity and LLM scores are utilized together, highlighting the importance of using Perplexity as well for example selection. Finally, the full `ActivePrune` algorithm, which integrates Perplexity, LLM Score, and Reweighting, achieves the highest ROUGE-L score of 30.85. This result con-

Table 3: Ablation of different components in `ActivePrune`.

| COMPONENT | | | ROUGE-L |
|---|---|---|---|
| Perplexity | LLM Score | Reweighting | |
| ✓ | | ✓ | 30.05 |
| ✓ | ✓ | | 30.34 |
| | ✓ | ✓ | 29.00 |
| ✓ | ✓ | ✓ | **30.85** |

firms that each component in `ActivePrune` contributes uniquely to the overall effectiveness.

## 6 CONCLUSION

In this work, we introduced `ActivePrune`, a novel plug-and-play method for enabling efficient active learning through data pruning using language models. Our approach follows a two-stage pruning process: a fast evaluation using perplexity scores from an $n$-gram language model, and a high-quality selection using quantized LLM. Additionally, we introduce a perplexity reweighting method that increases the diversity of the selected examples. Our experiments with diverse datasets and tasks demonstrate that `ActivePrune` effectively reduces the size of the unlabeled pool without compromising the selection quality of examples. Moreover, `ActivePrune` enables more efficient utilization of resources during active learning by reducing the computational cost associated with large unlabeled pools. Future work includes the application of `ActivePrune` on larger datasets to demonstrate its utility in enabling efficient active learning at scale.

**Limitations.** The implications of pruning the unlabeled pool on the privacy and fairness of the final trained models were not explored in this study. Data pruning may lead to disproportionate pruning of different subpopulations in the dataset, potentially leading to the insufficient representation of specific groups. To address these issues, it is essential to thoroughly evaluate explainability and fairness for models that undergo active learning after a data pruning procedure.

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

# A   PROOF OF PROPOSITION 1

*Proof.* Let $\mathcal{U} = \{x_i\}_{i=1}^N$ denote the unlabeled pool, and let $P_t(x_i)$ be the perplexity score of instance $x_i$ at iteration $t$. The labeled set at iteration $t$ is $\mathcal{L}_t$, with newly labeled instances $L_t \subseteq \mathcal{L}_t$. The adjustment factor for $x_i$ is defined as:

$$A_t(x_i) = \frac{1}{|L_t|} \sum_{l \in L_t} |P_t(x_i) - P_t(l)|.$$

The reweighted perplexity is updated according to:

$$P_{t+1}(x_i) = P_t(x_i) - \beta \cdot A_t(x_i), \tag{3}$$

where $\beta > 0$ is a hyperparameter controlling the adjustment magnitude. The probability of selecting $x_i$ at iteration $t$ is proportional to:

$$\pi_t(x_i) = \frac{w_t(x_i)}{\sum_{x_j \in \mathcal{U}} w_t(x_j)},$$

with $w_t(x_i) = f(P_t(x_i))$, and $f$ is a monotonically decreasing function since lower perplexity implies higher selection priority.

Let $S \subseteq \mathcal{U}$ represent an underrepresented subset, meaning the instances in $S$ have perplexity scores significantly different from those in $\mathcal{L}_t$, and these are not selected through the LLM quality scoring as well. Our goal is to show that for any $x_i \in S$, the sampling probability $\pi_t(x_i)$ increases over iterations, ensuring eventual selection.

Since $x_i \in S$ is underrepresented, there exists a constant $\delta > 0$ such that:

$$|P_t(x_i) - P_t(l)| \geq \delta, \quad \forall l \in \mathcal{L}_t.$$

Similarly, for instances $x_j \notin S$, which are similar to labeled instances, there exists a constant $\epsilon \geq 0$ (typically small) such that:

$$|P_t(x_j) - P_t(l)| \leq \epsilon, \quad \forall l \in \mathcal{L}_t.$$

Under the assumption that underrepresented instances are sufficiently different from labeled instances compared to other instances, we have $\delta > \epsilon$.

Therefore, the adjustment factor for $x_i \in S$ satisfies:

$$A_t(x_i) = \frac{1}{|L_t|} \sum_{l \in L_t} |P_t(x_i) - P_t(l)| \geq \delta. \tag{4}$$

Similarly, for $x_j \notin S$, the adjustment factor satisfies:

$$A_t(x_j) = \frac{1}{|L_t|} \sum_{l \in L_t} |P_t(x_j) - P_t(l)| \leq \epsilon. \tag{5}$$

Substituting equation 4 into equation 3, we obtain for $x_i \in S$:

$$P_{t+1}(x_i) \leq P_t(x_i) - \beta\delta.$$

Similarly, for $x_j \notin S$, substituting equation 5 into equation 3, we have:

$$P_{t+1}(x_j) \geq P_t(x_j) - \beta\epsilon.$$

Now, consider the difference in perplexity between $x_i \in S$ and $x_j \notin S$:

$$P_{t+1}(x_i) - P_{t+1}(x_j) = (P_t(x_i) - \beta A_t(x_i)) - (P_t(x_j) - \beta A_t(x_j))$$
$$= (P_t(x_i) - P_t(x_j)) - \beta \left( A_t(x_i) - A_t(x_j) \right).$$

Since $A_t(x_i) - A_t(x_j) \geq \delta - \epsilon$, we have:

$$P_{t+1}(x_i) - P_{t+1}(x_j) \leq (P_t(x_i) - P_t(x_j)) - \beta \left( \delta - \epsilon \right). \tag{6}$$

Define $\Delta = \beta \left( \delta - \epsilon \right) > 0$, since $\delta > \epsilon$ and $\beta > 0$.

Thus, the perplexity gap decreases by at least $\Delta$ per iteration:

$$P_{t+1}(x_i) - P_{t+1}(x_j) \leq (P_t(x_i) - P_t(x_j)) - \Delta. \tag{7}$$

Let $D_0 = P_0(x_i) - P_0(x_j)$. The number of iterations required for $x_i$ to have a perplexity less than or equal to that of $x_j$ is:

$$T_{ij} = \left\lceil \frac{D_0}{\Delta} \right\rceil. \tag{8}$$

Since there are a finite number of instances $x_j \notin S$, we can define:

$$T^* = \max_{x_j \notin S} T_{ij}. \tag{9}$$

At iteration $T^*$, for all $x_j \notin S$, we have:

$$P_{T^*}(x_i) \leq P_{T^*}(x_j).$$

Moreover, since the perplexity of $x_i$ decreases by at least $\beta\delta$ per iteration, from equation 4 and equation 3, the perplexity of $x_i$ after $T^*$ iterations satisfies: $P_{T^*}(x_i) \leq P_0(x_i) - \beta\delta T^*$. Similarly, for $x_j \notin S$, their perplexity satisfies: $P_{T^*}(x_j) \geq P_0(x_j) - \beta\epsilon T^*$. Thus, after $T^*$ iterations, $x_i \in S$ will have a perplexity not greater than any $x_j \notin S$.

An instance $x_i$ will be selected when its perplexity score ranks among the lowest $k$ in the unlabeled pool:

$$P_t(x_i) \leq P_t^{(k)}, \tag{10}$$

where $P_t^{(k)}$ is the $k$-th lowest perplexity score at iteration $t$. Since after $T^*$ iterations, $x_i$ has a perplexity lower than or equal to that of all $x_j \notin S$, and there are only a finite number of instances with perplexity less than or equal to $P_{T^*}(x_i)$, $x_i$ will eventually satisfy the selection condition equation 10.

Therefore, underrepresented instances $x_i \in S$ have larger adjustment factors, leading to a significant decrease in their reweighted perplexity scores over iterations. The perplexity gap between $x_i$ and other instances decreases at a rate of $\Delta = \beta(\delta - \epsilon) > 0$, resulting in $x_i$ eventually ranking among the lowest perplexity scores and being selected.

$\square$

## B  IMPLEMENTATION DETAILS

### B.1  ALGORITHM

We have extended the `ALToolbox`[2] Python package (Tsvigun et al., 2022a) to include functionality for `ActivePrune`. This enables compatibility with any dataset[3] and model[4] that is accessible on HuggingFace (Wolf et al., 2019). The `ActivePrune` procedure is presented in Algorithm 1. The ALToolbox repository is available under the MIT license. We release the code with the MIT license too and submit it as part of the supplementary material.

### B.2  RESOURCES

We utilized 48GB NVIDIA A6000 GPUs to conduct all our experiments on the cloud. Over the course of the project, from the initial trials to achieving the final results, the experiments used a total of approximately 3210 GPU hours.

---

[2]`https://github.com/AIRI-Institute/al_toolbox`

[3]`https://huggingface.co/datasets`

[4]`https://huggingface.co/models`

---

**Algorithm 1** `ActivePrune`

---

1: **Input:** Unlabeled pool $\mathcal{U}$, Budget $b$
2: **Output:** Labeled set $\mathcal{L}$
3: Initialize $\mathcal{L} = \emptyset$
4: $\mathbf{P} \leftarrow$ ComputePerplexityScores($\mathcal{U}$)          ▷ Compute perplexity for all unlabeled data
5: **while** $|\mathcal{L}| < b$ **do**          ▷ Begin active learning iterations until budget is exhausted
6:      $\mathcal{F} \leftarrow \emptyset$          ▷ Initialize filtered pool
7:      $\mathcal{P}_s \leftarrow$ SelectLowPerplexity($\mathbf{P}$)
8:      $\mathcal{F} \leftarrow \mathcal{F} \cup \mathcal{P}_s$          ▷ Add to filtered pool
9:      $\mathbf{Q} \leftarrow$ ComputeLLMDataQualityScores($\mathcal{U} \setminus \mathcal{P}_s$)
10:      $\mathcal{Q}_s \leftarrow$ SelectHighQuality($\mathbf{Q}$)
11:      $\mathcal{F} \leftarrow \mathcal{F} \cup \mathcal{Q}_s$          ▷ Add to filtered pool
12:      $S \leftarrow$ AcquisitionFunction($\mathcal{F}$)          ▷ Select most informative samples
13:      $\mathcal{L} \leftarrow \mathcal{L} \cup S$          ▷ Label and add samples to labeled set
14:      $\mathcal{U} \leftarrow \mathcal{U} \setminus S$          ▷ Remove labeled samples from pool
15:      ReweightPerplexity($\mathcal{U}, S$)          ▷ Adjust perplexity scores for diversity
16: **end while**
17: **procedure** REWEIGHTPERPLEXITY($\mathcal{U}, L, \beta$)
18:      **for all** $x_i \in \mathcal{U}$ **do**
19:          $A(x_i, L) \leftarrow \frac{1}{|L|} \sum_{l \in L} |P(x_i) - P(l)|$          ▷ Compute adjustment factor
20:          $P_{\text{new}}(x_i) \leftarrow P_{\text{old}}(x_i) - \beta . A(x_i, L)$          ▷ Update perplexity based on factor
21:      **end for**
22:      $\mathbf{P} \leftarrow$ Update all $P_{\text{new}}(x_i)$ for $x_i \in \mathcal{U}$
23: **end procedure**

---

### B.3 MODELS

Table 4: Summary of models used in the experiments, including their parameter count, architecture, use cases, and licenses.

| Model | Parameters | Architecture | Task | License |
|---|---|---|---|---|
| BART (Lewis et al., 2019) | 406M | Encoder-Decoder | ATS, NMT | MIT |
| RoBERTa-base (Liu et al., 2019) | 125M | Encoder-only | Classification | MIT |
| Gemma-2B (Team et al., 2024) | 2B | Decoder-only | LLM data quality scoring | Apache 2.0 |

### B.4 DATA STATISTICS

Table 5: The datasets used for experiments, their respective tasks, the number of training, validation, and test samples, and their licenses.

| Dataset | Task | Training | Validation | Test | License |
|---|---|---|---|---|---|
| **IMDB** | Sentiment Analysis | 25,000 | N/A | 25,000 | Non-commercial use only |
| **AG News** | Topic Classification | 120,000 | N/A | 7,600 | Non-commercial use only |
| **IT Domain** | Machine Translation (En-De) | 223,000 | 2,000 | 2,000 | Creative Commons CC0 |
| **AESLC** | Abstractive Text Summarization | 14,400 | 1,960 | 1,910 | CC BY-NC-SA 4.0 |

## C ADDITIONAL EXPERIMENTS

### C.1 MEDICAL DOMAIN DATASET

We conduct an experiment on a challenging medical domain dataset (from the multi-domains dataset Aharoni & Goldberg (2020)) for the En-De translation task, with 248,099 sentence pairs for training and 2000 for testing. We use the BART-base variant trained on the WMT14 En-De out-of-domain parallel corpora and then use the medical dataset for the domain adaptation. The SacreBLEU and COMET scores for the final AL iteration shown in the table below show that ActivePrune performs better than other strategies consistently.

| | ActivePrune | Random | ASK-LLM | Perplexity | UPS |
|---|---|---|---|---|---|
| **SacreBLEU** | **36.39** | 35.22 | 35.05 | 35.96 | 35.60 |
| **COMET** | **45.27** | 44.38 | 43.11 | 44.63 | 44.12 |

Table 6: Performance comparison of `ActivePrune` with other pruning methods on the medical domain dataset.

## C.2 SummaC-ZS Scores for Summarization Task

To fully capture output quality in the summarization task and assess the factual consistency of the generated summaries in relation to the original documents, we utilize SummaC (Laban et al., 2022), a state-of-the-art model for detecting inconsistencies. We compute the SummaC-ZS scores for all strategies. The scores shown in Table 7 demonstrate that ActivePrune shows better performance compared to other pruning methods consistently across all iterations.

| Data Pruning Strategy | 1 | 2 | 3 | 4 | 5 |
|---|---|---|---|---|---|
| ASK-LLM | 17.65 | 17.05 | 18.00 | 18.10 | 18.78 |
| Perplexity | 16.40 | 15.33 | 17.92 | 18.80 | 17.05 |
| UPS | 15.76 | 17.25 | 16.40 | 17.85 | 17.48 |
| Random | 15.07 | 17.08 | 17.31 | 16.51 | 17.75 |
| ActivePrune | **15.47** | **17.89** | **18.67** | **19.25** | **20.25** |

Table 7: SummaC-ZS scores comparison across different pruning methods.

## C.3 Empirical analysis of perplexity reweighting

To empirically demonstrate the promotion of underrepresented instances through perplexity reweighting in `ActivePrune`, we compute the diversity of the selected batch in each iteration, using the diversity score presented in Dor et al. (2020). We compute the diversity of the selected batch with and without perplexity reweighting procedure for the summarization task with the AESLC dataset and IDDS strategy. For the distance measure between two examples, we use the difference in the perplexity values of the instances. We find that the perplexity reweighting procedure consistently leads to higher diversity in each iteration compared to no reweighting, suggesting that it successfully leads to the inclusion of the underrepresented instances in the selected subset. The results are shown in Table 8.

| | 1 | 2 | 3 | 4 | 5 |
|---|---|---|---|---|---|
| **Without perplexity reweighting** | 9.74 | 12.39 | 12.55 | 8.73 | 10.10 |
| **With perplexity reweighting** | 9.74 | **12.5** | **12.89** | **12.42** | **11.90** |

Table 8: Comparison of the diversity across iterations with and without perplexity reweighting.

## C.4 Combining ActivePrune with Contrastive Active Learning

Contrastive Active Learning (CAL) (Margatina et al., 2021b) is a state-of-the-art AL acquisition method that balances uncertainty and diversity by selecting contrastive examples – data points similar in the model feature space but with maximally different predictive likelihoods – demonstrating superior performance across multiple natural language understanding tasks and datasets. We conudct an experiment combining ActivePrune with Contrastive Active Learning (CAL) on the IMDB dataset for the classification task. The results, shown in Table 9, demonstrate that ActivePrune can effectively complement state-of-the-art AL methods like CAL to achieve higher performance.

| Data Pruning Strategy | 1 | 2 | 3 | 4 | 5 |
|---|---|---|---|---|---|
| Perplexity | 91.57 | 92.40 | 92.60 | 92.87 | 92.76 |
| Random | 91.93 | 92.09 | 92.51 | 92.47 | 92.80 |
| UPS | 88.24 | 91.86 | 92.10 | 92.59 | 92.83 |
| ASK-LLM | 90.71 | 90.53 | 89.39 | 90.57 | 90.37 |
| ActivePrune | **92.03** | **92.44** | **92.68** | **93.04** | **93.95** |

Table 9: F1-score across different iterations when combining data pruning methods with Contrastive Active Learning.

## C.5 EVALUATION WITH BERT

ActivePrune is an AL-strategy agnostic method, and its performance is expected to generalize across various models. To verify this, we conduct an experiment on an additional model, BERT, for the sentiment analysis task on the IMDB dataset. The results, summarized in Table 10, show that ActivePrune delivers consistent F1-score improvements over the other strategies across most iterations on BERT as well.

| Data Pruning Strategy | 1 | 2 | 3 | 4 | 5 |
|---|---|---|---|---|---|
| Perplexity | 87.91 | 88.71 | 89.33 | 89.22 | 89.99 |
| Random | 87.93 | 88.56 | 89.66 | 89.35 | 89.67 |
| UPS | 84.50 | 87.32 | 88.11 | 89.77 | 89.60 |
| ASK-LLM | 86.19 | 85.49 | 87.24 | 86.63 | 87.01 |
| ActivePrune | 88.37 | 88.74 | 88.42 | 90.06 | 90.29 |

Table 10: F1-score across different iterations with BERT model on the classification task with IMDB dataset.

## C.6 K-MEANS CLUSTERING AS A DATA PRUNING BASELINE

We conduct an experiment to compare ActivePrune with K-Means clustering as a lightweight data pruning baseline. We follow the standard procedure of clustering the embedding representation and then including representative examples from each cluster in the final selected subset. The F1 scores are shown in the in Table 11 for IMDB dataset with LC strategy. We've also included the scores of other strategies as well for easier comparison. We find that ActivePrune performs better than K-means consistently across all iterations.

| Data Pruning Strategy | 1 | 2 | 3 | 4 | 5 |
|---|---|---|---|---|---|
| Random | 91.73 | 92.59 | 92.63 | 92.85 | 93.10 |
| ASK-LLM | 90.13 | 90.26 | 88.30 | 90.15 | 90.33 |
| Perplexity | 91.71 | 92.08 | 92.18 | 92.74 | 92.79 |
| UPS | 88.56 | 91.89 | 92.47 | 92.75 | 93.21 |
| K-Means | 90.85 | 91.83 | 92.22 | 91.94 | 92.38 |
| ActivePrune | **91.97** | **92.22** | **92.79** | **92.87** | **93.25** |

Table 11: Comparison of data pruning methods with K-Means clustering

# D QUALITATIVE COMPARISON

We present a qualitative comparison of perplexity and LLM data quality scores for the IT domain En-De translation dataset. Table 12 contains examples with high percentiles of perplexity scores, representing the examples with the lowest values of perplexity. Table 13 shows examples with a high percentile of LLM scores, representing the examples with the highest values of data quality scores.

Table 12: A selection of translation examples from the IT dataset with high perplexity percentiles and varying LLM quality percentiles.

| PERPLEXITY PERCENTILE | LLM SCORE PERCENTILE | TEXT (ENGLISH) | TRANSLATION (GERMAN) |
|---|---|---|---|
| 99.86 | 94.47 | With this option you can turn the & kmix; main window content by 90 degrees. Sliders, texts and everything else (if applicable) is rotated. There are some exclusions to this rule, most notably the menubar, the mixer selector (if shown at all), the panning slider and the multiple-choice selectors. | Mit dieser Einstellung können Sie den Inhalt des & kmix;-Hauptfensters um 90 Grad drehen / Schieberegler, Texte und vieles mehr wird gedreht. Davon ausgeschlossen sind & eg; die Menüleiste, die Mixer-auswahl (wenn vorhanden), der Rechts-Links-Abgleich und die Mehrfachauswahlen. |
| 99.71 | 88.45 | One is used for text completion in the Location Toolbar text entry box. To clear this right click on the text entry box and select Clear History. | Einer wird für die Vervollständigung von Einträgen im Eingabefeld der Adressleiste benutzt. Um diese Einträge zu löschen, klicken Sie mit der rechten Maustaste auf das Texteingabefeld und wählen Alles löschen aus. |
| 99.84 | 79.84 | Build a temporary sandbox and download/install the necessary packages and debug symbols in there; without this option it assumes that the necessary packages and debug symbols are already installed in the system. The argument points to the packaging system configuration base directory; if you specify "system", it will use the system configuration files, but will then only be able to retrace crashes that happened on the currently running release. | Erstellt eine temporäre Sandbox, lädt die benötigten Pakete und Symbole zur Fehlerdiagnose herunter und installiert diese in die Sandbox; ohne diese Option wird davon ausgegangen, dass die benötigten Pakete und Symbole zur Fehlerdiagnose bereits im System installiert sind.Der Parameter verweist auf das Stammverzeichnis des Paketsystemkonfigurationsverzeichnisses; wenn Sie »system« angeben werden die Systemkonfigurationsdateien verwendet, allerdings sind Sie dann nur in der Lage Abstürze in der aktuell laufenden Veröffentlichung zurück zu verfolgen. |
| 99.73 | 72.93 | Color-blindness simulation | Simulation für Farbenblindheit |
| 99.67 | 69.30 | akonadictl found and usable | akonadictl gefunden und benutzbar |
| 99.93 | 25.23 | _Add Bookmark... | Lesezeichen _hinzufügen ... |
| 99.96 | 10.50 | File '%1 'not found! | Datei, %1' kann nicht gefunden werden. |
| 99.91 | 6.77 | & Audio Alarm Template | & Audio-Erinnerungsvorlage@action |
| 99.86 | 3.69 | Local Scope | Lokaler Gültigkeitsbereich |
| 99.88 | 1.24 | Auto Next | Automatisch zum nächsten |

Table 13: A selection of translation examples from the IT dataset with high LLM quality scores and varying perplexity scores.

| PERPLEXITY PERCENTILE | LLM SCORE PERCENTILE | TEXT (ENGLISH) | TRANSLATION (GERMAN) |
|---|---|---|---|
| 96.53 | 99.80 | The 'Zoom to 100%' command resizes the video area to the original video size of the current file. | ‚Äû Zoomfaktor 100 %‚Äú zeigt das Video in Originalgröße an. |
| 82.54 | 99.98 | Mark this check box to convert the URLs referencing other ODF files to PDF files with the same name. | Aktivieren Sie diese Option, um URLs, die von anderen ODF-Dateien auf PDF-Dateien verweisen, mit dem gleichen Namen zu konvertieren. |
| 65.46 | 99.69 | Designed to be the successor of the MP3 format, AAC generally achieves better sound quality than MP3 at many bit rates. | Wird als Nachfolger des MP3-Formats angesehen. AAC erreicht generell bessere Klangqualität als MP3 bei niedrigeren Bitraten. |
| 51.42 | 99.65 | To get a list of all Kanji with a certain number of strokes, enter that number in the text-entry in the toolbar as "S:4". | Um eine Liste aller Kanji mit einer bestimmten Anzahl von Strichen zu bekommen, geben Sie diese Anzahl in das Texteingabefeld in der Werkzeugleiste als S:4 ein. |
| 36.20 | 99.64 | Remove the selected widget, returns the index of the removed widget or -1 if no such widget was found. | Entfernt das ausgewählte Bedienelement, gibt den Index des entfernten Bedienelements zurück oder -1, falls kein Bedienelement gefunden wurde. |
| 33.61 | 99.60 | SEEK-END - Set position to end-of-file plus offset. (To move to a position before the end-of-file, you need to pass a negative value in offset.) | SEEK-END - Setzt die Position ans Ende der Datei plus offset. (Um zu einer Position vor EOF zu gelangen, übergeben Sie in offset einen negativen Wert.) |
| 27.69 | 99.91 | Error reading the keyboard layout; the default number keypad will be created instead. You can choose another keyboard layout in the preferences dialog. | Fehler beim Lesen der Tastaturbelegung! Stattdessen wird das normale Nummernfeld erzeugt. Sie können eine andere Tastaturbelegung im Dialog ‚Äû Einstellungen‚Äú wählen. |
| 15.20 | 99.86 | Use –filter='exclude. rsync-filter 'filter rule | Benutze die –filter=‚Äö exclude .rsync-filter‚Äô Filterregel |
| 13.82 | 99.59 | Cut off after specified number. Use -1 for infinite number of entries. | Nach angegebener Anzahl abschneiden. -1 bedeutet unbegrenzt. |
| 2.41 | 99.75 | Only reuses an instance with the specified PID (Process ID). Used with the –use option. | kate; benutzt eine existierende Instanz nur dann, wenn diese die angegebene PID (Process ID) hat. Diese Option wird zusammen mit der Option –use benutzt. |

