# OpenReview forum: "Language Model-Driven Data Pruning Enables Efficient Active Learning"
_ICLR.cc/2025/Conference — Submitted to ICLR 2025_

### Official Review · Reviewer_doHz · 2024-11-01

**Soundness:** 2
**Presentation:** 3
**Contribution:** 2
**Rating:** 5
**Confidence:** 4

**Summary:**

The paper introduces ActivePrune, a method designed to enhance the efficiency of active learning techniques in NLP. In active learning, instances from an unlabeled pool are selected for labeling, which can be challenging when the pool is large. ActivePrune addresses this issue through pruning: it utilizes perplexity scoring and LLM filtering to reduce the initial pool to a more manageable size. Subsequently, active learning algorithms are applied to this refined pool. Experiments conducted on four NLP tasks demonstrate that ActivePrune is more efficient than existing methods.

**Strengths:**

The strengths of the paper are presented below:

- S1) The problem studied is interesting, as it is computationally challenging to evaluate the LLM's uncertainty on a large pool of examples.

- S2) Experiments conducted on the large IT dataset show that ActivePrune is more efficient than other existing uncertainty-based methods.

- S3) The paper is clearly organized and easy to follow. Additionally, Section 3.4 provides a solution for addressing underrepresented instances during pool pruning.

**Weaknesses:**

The weaknesses of the paper are presented below:

- W1) **Evaluation**: The experimental evaluation lacks depth. For instance, ActivePrune compares itself to other costly LLM-based pruning techniques but does not include comparisons with lightweight methods, such as k-means clustering or coreset selection. These experiments are essential to assess the effectiveness of the approach.
- W2) **Soundness**: ActivePrune achieves improved efficiency through the use of an n-gram model for perplexity scoring and a quantized 2B LLM for filtering. However, such a solution is not sophisticated and lack robustness. For example, in domain-specific datasets, such as medical or finance, or in complex tasks, such as math reasoning, these models might be limited. It is suggested that authors evaluate their approach in more challenging tasks.
- W3) Furthermore, the approach does not account for the downstream LLM. For example, it may prune the same instances regardless of whether a medical or finance LLM is used, omitting to consider their respective knowledge of the task.

**Questions:**

Please refer to the comments above. Here are some additional questions and comments:
- Q1) Section 3.3: Could you clarify why you chose a prompt-based approach to evaluate perplexity instead of directly assessing it based on the LLM's logits for the sequence?
- Q2) Section 4.2: Why do you select only one model per task? Would it be possible to evaluate multiple models on the same task?
- Q3) Have you considered using in-context learning in addition to training models on the selected examples for active learning?
- Q4) Section 3.2: I recommend moving some details from this paragraph to another section, as they are not directly related to the proposed method.

---

> ### Author Response · Authors · 2024-11-20
> **Response to Reviewer doHz (Part 1)**
>
> Thank you for your thoughtful and detailed feedback! We appreciate your suggestions, and we have addressed each concern below:
>
> > W1) Evaluation: The experimental evaluation lacks depth. For instance, ActivePrune compares itself to other costly LLM-based pruning techniques but does not include comparisons with lightweight methods, such as k-means clustering or coreset selection. These experiments are essential to assess the effectiveness of the approach.
>
> Thank you for the feedback. Unlabeled pool subsampling (UPS) is a **lightweight** and **strong** baseline for data pruning in AL, and we included this in comparison with ActivePrune for all the tasks that we evaluated on. Table 1 shows that ActivePrune outperforms unlabeled pool subsampling consistently across all iterations.
>
> Coreset selection is typically used as an AL method instead of a data pruning method for reducing the size of unlabeled pool, thus we included it as an AL baseline. It was tested on the classification tasks on IMDB and AG-News datasets, and the results are shown in Table 1 in the paper.
>
> As suggested, we have now conducted an additional experiment with **k-means clustering** as the data pruning method. We follow the standard procedure of clustering the embedding representation and then including representative examples from each cluster in the final selected subset. The F1 scores are shown in the table below for IMDB dataset with LC strategy. We’ve also included the scores of other strategies as well in the Table below for easier comparison. We find that **ActivePrune performs better than K-means consistently across all iterations**.
>
>
>
> | Data Pruning Method  | Iteration |  |  |  |  |
> |---|---|---|---|---|---|
> | | 1 | 2 | 3 | 4 | 5 |
> | Random        | 91.73      | 92.59      | 92.63      | 92.85      | 93.10      |
> | ASK-LLM       | 90.13      | 90.26      | 88.30      | 90.15      | 90.33      |
> | Perplexity    | 91.71      | 92.08      | 92.18      | 92.74      | 92.79      |
> | UPS           | 88.56      | 91.89      | 92.47      | 92.75      | 93.21      |
> | K-Means           | 90.85      | 91.83      | 92.22      | 91.94      |92.38       |
> | ActivePrune   | **91.97**  | **92.22**  | **92.79**  | **92.87**  | **93.25**  |
>
>
> > W2) Soundness: ActivePrune achieves improved efficiency through the use of an n-gram model for perplexity scoring and a quantized 2B LLM for filtering. However, such a solution is not sophisticated and lack robustness. For example, in domain-specific datasets, such as medical or finance, or in complex tasks, such as math reasoning, these models might be limited. It is suggested that authors evaluate their approach in more challenging tasks.
>
> Thank you for the excellent suggestion, We have now conducted an evaluation on the medicine domain (from the multi-domains dataset [1]) for the En -> De translation task, with 248,099 sentence pairs for training and 2000 for testing. We use the BART-base variant (with 139 million parameters) trained on the WMT14 En-De out-of-domain parallel corpora and then use the medicine dataset for the domain adaptation. The SacreBLEU and COMET scores for the final AL iterations presented in the table below demonstrate that **ActivePrune performs better than other strategies consistently**:
>
>
> |  | ActivePrune | Random | ASK LLM | Perplexity | UPS |
> |---|---|---|---|---|---|
> | SacreBLEU | **36.39** | 35.22 | 35.05 | 35.96 | 35.60 |
> | COMET | **45.27** | 44.38 | 43.11 | 44.63 | 44.12 |
>
> > W3) Furthermore, the approach does not account for the downstream LLM. For example, it may prune the same instances regardless of whether a medical or finance LLM is used, omitting to consider their respective knowledge of the task.
>
> We would like to highlight that the examples that are selected in each AL iteration by the downstream model are passed as inputs to the perplexity reweighting procedure (Figure 1). The reweighting procedure uses those for computation and then systematically brings forward underrepresented instances for selection in subsequent iterations. Therefore, if a medical or finance LLM is used as the downstream model, the instances selected in each iteration would provide the necessary information about the selection preference of that LLM, and thus **actively influence the pruning procedure in each iteration**.

---

> ### Author Response · Authors · 2024-11-20
> **Response to Reviewer doHz (Part 2)**
>
> > Q1) Section 3.3: Could you clarify why you chose a prompt-based approach to evaluate perplexity instead of directly assessing it based on the LLM's logits for the sequence?
>
> We would like to clarify that we *did not* use a prompt-based approach to evaluate perplexity. Perplexity was computed through the KenLM n-gram language model. In contrast, the LLM data quality scores were computed using the LLM’s logits output for a particular prompt.
>
> > Q2) Section 4.2: Why do you select only one model per task? Would it be possible to evaluate multiple models on the same task?
>
> ActivePrune is an AL-strategy agnostic method, and its performance is expected to generalize across various models. We have now performed an experiment on an additional model, BERT, for the sentiment analysis task on the IMDB dataset. The results, summarized in the table below, show that **ActivePrune delivers consistent F1-score improvements over the other strategies across most iterations on BERT as well**.
>
> | Data Pruning Method |  Iteration |  |  |  |  | |
> |---|---|---|---|---|---|---|
> |   | 1 | 2 | 3 | 4 | 5 |
> | Perplexity | 87.91 | 88.71 | 89.33 | 89.22 | 89.99 |
> | Random | 87.93 | 88.56 | 89.66 | 89.35 | 89.67 |
> | UPS | 84.50 | 87.32 | 88.11 | 89.77 | 89.60 |
> | ASK-LLM | 86.19 | 85.49 | 87.24 | 86.63 | 87.01 |
> | ActivePrune | 88.37 | 88.74 | 88.42 | 90.06 | 90.29 |
>
> > Q3) Have you considered using in-context learning in addition to training models on the selected examples for active learning?
>
> We did not consider in-context learning in this work, as we followed the standard active learning protocol that includes fine-tuning the model on the examples labeled in each iteration. However, this could be an interesting direction to consider when LLM is used in the AL procedure, and could further improve the efficiency of the AL process. We have mentioned this in the future work.
>
> > Q4) Section 3.2: I recommend moving some details from this paragraph to another section, as they are not directly related to the proposed method.
>
> Thank you for the suggestion. We have made this change in the revised version.
>
>
>
> We sincerely thank you for your insightful feedback, which has helped us strengthen our work. We believe that the revisions effectively address your concerns.
>
>
> References:
>
> [1] https://aclanthology.org/2020.acl-main.692.pdf

---

> ### Comment · Reviewer_doHz · 2024-11-23
>
> Thank you to the authors for their thoughtful response and the additional experiments. I appreciate the clarifications provided, and several of my initial questions have been addressed. As a result, I have increased my score accordingly.
>
> However, I would like to recommend that the authors further refine the steps for calculating perplexity (Section 3.2) and LLM scoring (Section 3.3). It would be beneficial to ensure these steps are more robust across different datasets, particularly in terms of domain variability and task complexity. The new experiments provided are promising, but expanding the evaluation to consider a broader range of conditions would strengthen the overall work.

---

> > ### Author Response · Authors · 2024-11-23
> >
> > Thank you for your response! We’re glad that the clarification and additional experiments addressed your concerns. We’ve included the new experimental results in the paper and will further clarify how the perplexity and LLM scoring procedure is robust across different domains and tasks.
> >
> > If there are any remaining concerns regarding the paper that you would like to be addressed, please let us know and we will attempt to address them.

---

> > > ### Author Response · Authors · 2024-11-26
> > >
> > > We have now provided further clarification on the robustness of perplexity and the LLM scoring procedure across various domains in Sections 3.2 and 3.3 (changes highlighted in blue). The additional experimental results provided earlier are also included in the appendix (C1 - C6).
> > >
> > > If there are any remaining concerns that you would like to be addressed, please let us know and we will attempt to address them.

---

### Official Review · Reviewer_WT6c · 2024-11-06

**Soundness:** 3
**Presentation:** 3
**Contribution:** 2
**Rating:** 6
**Confidence:** 4

**Summary:**

The authors propose a data pruning method for active learning (AL) settings that discards instances from the unlabeled pool using a quantized large language model (LLM). Experiments across four datasets from machine translation, summarization, and classification demonstrate that the proposed method outperforms several baselines and is computationally faster than other LLM-based pruning methods.

**Strengths:**

- The paper is well-written, and the method is clearly explained.
- The proposed idea is straightforward and appears to be computationally faster than similar state-of-the-art LLM-based methods.
- The method works in both classification and text generation tasks.

**Weaknesses:**

- The related work section is missing several active learning works (please see below).
- Improvements seem marginal.
- Metrics like BLEU and ROUGE do not fully capture output quality in text generation experiments. The authors should consider using a wider range of metrics to provide a more comprehensive evaluation.
- Evaluation uses a limited number of datasets.

### Missing References
1. Xun Deng, Wenjie Wang, Fuli Feng, Hanwang Zhang, Xiangnan He, and Yong Liao. 2023. Counterfactual Active Learning for Out-of-Distribution Generalization.
2. Katerina Margatina, Giorgos Vernikos, Loïc Barrault, and Nikolaos Aletras. 2021. Active Learning by Acquiring Contrastive Examples.
3. Michelle Yuan, Hsuan-Tien Lin, and Jordan Boyd-Graber. 2020. Cold-start Active Learning through Self-supervised Language Modeling.
4. Shujian Zhang, Chengyue Gong, Xingchao Liu, Pengcheng He, Weizhu Chen, and Mingyuan Zhou. 2022. ALLSH: Active Learning Guided by Local Sensitivity and Hardness.
5. Michalis Korakakis, Andreas Vlachos, and Adrian Weller. 2024. ALVIN: Active Learning Via INterpolation.
6. Liat Ein-Dor, Alon Halfon, Ariel Gera, Eyal Shnarch, Lena Dankin, Leshem Choshen, Marina Danilevsky, Ranit Aharonov, Yoav Katz, and Noam Slonim. 2020. Active learning for BERT: An empirical study.

**Questions:**

- How does the method impact model generalisation in out-of-distribution settings? CounterAL and ALVIN show that many AL methods struggle with OOD generalisation, does this method address that?
- The authors claim that the method promotes underrepresented instances but do not demonstrate that empirically. Metrics such as those defined in Ein-Dor et al. (2020) could help assess the types of instances chosen. Furthermore, if the proposed method indeed identifies underrepresented instances, this could also enhance the method’s effectiveness in OOD settings, as mentioned above.
- It would have been interesting to see how the proposed method performs when combined with state-of-the-art AL methods, such as CAL.
- How does the method perform when used to train other model architectures, such as BERT?

---

> ### Author Response · Authors · 2024-11-20
> **Response to Reviewer WT6c**
>
> Thank you for your thoughtful review and feedback! We appreciate the comments regarding our work. The specific concerns are addressed below:
>
> > W1: The related work section is missing several active learning works (please see below).
>
> Thank you. We’ve included these in the related works section now.
>
> > W2: Improvements seem marginal.
>
> We would like to emphasize that the performance improvement with ActivePrune is **statistically significant** across all datasets used in our experiments (line 400-402), including IT, AESLC, IMDB, and AG News. In addition to the better. Furthermore, ActivePrune offers substantial computational efficiency, achieving a superior accuracy-efficiency trade-off compared to existing pruning strategies. This balance between accuracy and efficiency is a key contribution of our approach.
>
> > W3: Metrics like BLEU and ROUGE do not fully capture output quality in text generation experiments. The authors should consider using a wider range of metrics to provide a more comprehensive evaluation.
>
> Thank you for this suggestion. In addition to BLEU and ROUGE, we have now computed COMET scores for translation (see experiments on the Medical dataset below) and SummaC [1] for the summarization task to provide a more comprehensive evaluation. These results demonstrate that **ActivePrune shows higher SummaC and COMET scores** compared to other strategies consistently across iterations.
>
> **SummaC scores for summarization task**:
>
> | Data Pruning Method  | Iteration |  |  |  |  |
> |---|---|---|---|---|---|
> |  | 1 | 2 | 3 | 4 | 5 |
> | ASK-LLM | 17.65 | 17.05 | 18.00 | 18.10 | 18.78 |
> | Perplexity | 16.40 | 15.33 | 17.92 | 18.80 | 17.05 |
> | UPS | 15.76 | 17.25 | 16.40 | 17.85 | 17.48 |
> | Random | 15.07 | 17.08 | 17.31 | 16.51 | 17.75 |
> | ActivePrune | 15.47 | **17.89** | **18.67** | **19.25** | **20.25** |
>
>
>
>
> > W4: Evaluation uses a limited number of datasets.
>
> We conducted our evaluation on four diverse domain datasets and tasks: the IT domain dataset for translation, AESLC for summarization, IMDB for sentiment analysis, and the AG News dataset for topic classification.
>
> We have now conducted an experiment on an additional medicine domain dataset as well (from the multi-domains dataset [2]) for the En -> De translation task, with 248,099 sentence pairs for training and 2000 for testing. We use the BART-base variant trained on the WMT14 En-De out-of-domain parallel corpora and then use the medicine dataset for the domain adaptation. The SacreBLEU and COMET scores for the final AL iterations shown in the table below show that **ActivePrune performs better than other strategies consistently**:
>
> |  | ActivePrune | Random | ASK LLM | Perplexity | UPS |
> |---|---|---|---|---|---|
> | SacreBLEU | **36.39** | 35.22 | 35.05 | 35.96 | 35.60 |
> | COMET | **45.27** | 44.38 | 43.11 | 44.63 | 44.12 |

---

> ### Author Response · Authors · 2024-11-20
> **Response to Reviewer WT6c (Part 2)**
>
> > Q1: How does the method impact model generalisation in out-of-distribution settings? CounterAL and ALVIN show that many AL methods struggle with OOD generalisation, does this method address that?
>
> This is an excellent question. ActivePrune’s two-stage pruning process followed by perplexity reweighting is designed to retain the informative as well as the underrepresented instances in the pruned dataset, as theoretically demonstrated in the proof in Appendix A. This ensures that strong AL methods can reliably select examples from the pruned dataset that are important for in-domain and out-of-domain generalization.
>
> ALVIN conducts intra-class interpolations between examples from under-represented and well-represented groups to create anchor points and demonstrates SOTA performance on OOD generalization. Given that our method already ensures the representation of the underrepresented and well-represented groups, **combining ActivePrune with methods like ALVIN would certainly enable efficiency while retaining their strong performance**.
>
> Lastly, we also tested ActivePrune in a domain adaptation setting on the IT dataset. The BART-based model, trained on the WMT14 En-De out-of-domain parallel corpora, was fine-tuned through AL on the in-domain IT dataset. The results in Table 1 show significant performance improvements with ActivePrune over other pruning methods, which indicates that the proposed method can reliably be used in a domain adaptation setting.
>
> > Q2: The authors claim that the method promotes underrepresented instances but do not demonstrate that empirically. Metrics such as those defined in Ein-Dor et al. (2020) could help assess the types of instances chosen. Furthermore, if the proposed method indeed identifies underrepresented instances, this could also enhance the method’s effectiveness in OOD settings, as mentioned above.
>
> Thank you for this suggestion. In the proof presented in Appendix A, we theoretically analyze the inclusion of underrepresented instances and demonstrate that their sampling probability increases over iterations, leading to their eventual selection.
>
> We have now conducted an experiment to empirically demonstrate the promotion of underrepresented instances by computing the diversity score suggested in Section 5 by Ein-Dor et al. [3].  We compute the diversity of the selected batch in each iteration with and without perplexity reweighting procedure for the summarization task with the AESLC dataset. For the distance measure (d_i) between two examples in the formula provided by Ein-Dor, we use the difference in the perplexity values of the instances. We find that **perplexity reweighting procedure consistently leads to higher diversity in each iteration compared to no reweighting**. The results are shown in the table below:
>
> |  | Iteration 1 | Iteration 2 | Iteration 3 | Iteration 4 | Iteration 5 |
> |---|---|---|---|---|---|
> | Without PPL reweighting | 9.74 | 12.39 | 12.55 | 8.73 | 10.10 |
> | With PPL reweighting | 9.74 | **12.5** | **12.89** | **12.42** | **11.90** |
>
>
> > Q3: It would have been interesting to see how the proposed method performs when combined with state-of-the-art AL methods, such as CAL.
>
> Thank you for the insightful suggestion. We have now conducted experiments combining ActivePrune with Contrastive Active Learning (CAL) on the IMDB dataset for the classification task. The results, shown in the table below, demonstrate that **ActivePrune can effectively complement state-of-the-art AL methods like CAL to achieve higher performance**.
>
> | Data Pruning Method  | Iteration |  |  |  |  |
> |---|---|---|---|---|---|
> | | 1 | 2 | 3 | 4 | 5 |
> | Perplexity | 91.57 | 92.40 | 92.60 | 92.87 | 92.76 |
> |  Random | 91.93 | 92.09 | 92.51 | 92.47 | 92.80 |
> |  UPS | 88.24 | 91.86 | 92.10 | 92.59 | 92.83 |
> |  ASK-LLM | 90.71 | 90.53 | 89.39 | 90.57 | 90.37 |
> |  ActivePrune | **92.03** | **92.44** | **92.68** | **93.04** | **93.95** |
>
>
> > Q4: How does the method perform when used to train other model architectures, such as BERT?
>
> We have now conducted an experiment with BERT, on the task of sentiment analysis with IMDB dataset. The results, summarized in the table below, show that **ActivePrune delivers consistent F1-score improvements over the other strategies** across most iterations on BERT as well.
>
> |  Data Pruning Method | Iteration | |  |  |  |  |
> |---|---|---|---|---|---|---|
> |   | 1 | 2 | 3 | 4 | 5 |
> | Perplexity | 87.91 | 88.71 | 89.33 | 89.22 | 89.99 |
> | Random | 87.93 | 88.56 | 89.66 | 89.35 | 89.67 |
> | UPS | 84.50 | 87.32 | 88.11 | 89.77 | 89.60 |
> | ASK-LLM | 86.19 | 85.49 | 87.24 | 86.63 | 87.01 |
> | ActivePrune | 88.37 | 88.74 | 88.42 | 90.06 | 90.29 |
>
>
>
>
> **References:**
>
> [1] https://arxiv.org/abs/2111.09525
>
> [2] https://arxiv.org/abs/2004.02105
>
> [3] https://aclanthology.org/2020.emnlp-main.638.pdf

---

> > ### Comment · Reviewer_WT6c · 2024-11-25
> >
> > Dear Authors,
> >
> > Thank you for your detailed response and the additional experiments addressing my comments. After reviewing these, I have decided to update my score accordingly.

---

> > > ### Author Response · Authors · 2024-11-25
> > >
> > > Thank you for your response! We’re glad to hear that the additional experiments addressed your concerns.

---

### Official Review · Reviewer_WdAB · 2024-11-07

**Soundness:** 3
**Presentation:** 2
**Contribution:** 2
**Rating:** 6
**Confidence:** 3

**Summary:**

This paper introduces ActivePrune a Data Pruning method for Active Learning. The method comes in 3 parts: a perplexity filter, a LLM model for quality selection, and perplexity reweighting for diversity. First, the perplexity filter reduces the unlabeled pool to top k examples. Second, the quantized LLM chooses quality examples by being prompted a yes or no questions. Lastly, the perplexity reweight formula  updates the perplexity scores in light of new data to increases diversity. The reweighting formula is also the main novelty of the paper. The authors claim that this method can speed up the training process and also improves task performances.

**Strengths:**

The writing is easy to follow an understand. All 3 stages of the methods are clearly stated and it is very easy to understand their mechanisms.

The diversity claim through perplexity is explained clearly with proofs.

Leveraging LLM is a good way to improve the quality selected examples.

The problem is non trivial as data size can greatly affect the training process.

**Weaknesses:**

A contradictory stance on the perplexity score: examples with the lowest perplexity scores are chosen to improve quality but then examples with high initial perplexity are then chosen also. The perplexity reweighting method does not target quality examples but only diversify the perplexity of the chosen examples.

Unclear why the perplexity - LLM pipeline is needed: Sachdeva et al., 2024 stated that the usage of LLM instead of perplexity to select quality data is because LLM is superior. The author has not justify their decision to use both methods in contrary to Sachdeva et al., 2024

Unjustified usage of perplexity for diversity: Based on figure 4, the selected high perplexity examples are low quality examples.

Figure 4 seems to suggests the Ask-LLM would be a better candidate as the chosen samples have higher quality and the mass of the selection examples are more spread out on the perplexity tail end compare to ActivePrune.

The speed up baseline is unreasonable: ActivePrune uses both perplexity and LLM but the speed up is 100x compare to methods that only use LLM or perplexity.

**Questions:**

Are the selected high perplexity examples actually good for training?

Does the competitiveness for speed go away when the baselines also use quantized LLM?

---

> ### Author Response · Authors · 2024-11-20
> **Response to Reviewer WdAB**
>
> Thank you for your thoughtful and detailed feedback! We appreciate the comments regarding the soundness and importance of the presented work. Addressing specific concerns and questions below:
>
> > A contradictory stance on the perplexity score: examples with the lowest perplexity scores are chosen to improve quality but then examples with high initial perplexity are then chosen also. The perplexity reweighting method does not target quality examples but only diversify the perplexity of the chosen examples.
>
> > Unjustified usage of perplexity for diversity: Based on figure 4, the selected high perplexity examples are low quality examples.
>
> Thank you for your feedback, We would like to clarify the perplexity selection procedure in ActivePrune:
> - Two-Stage Selection:
>     - In the first stage of the pipeline involving perplexity filtering, we intentionally focus on selecting the **prototypical and easier-to-learn** examples, which improves overall training stability and quality.
>     - In the second stage, examples with higher perplexity scores are selectively included based on their LLM data quality scores. This step focuses on identifying **challenging but valuable examples**. The reason for only selectively picking high PPL examples in the second stage is that the noisier examples that are detrimental to the overall training also have higher perplexity (L134-136, Section 3.2 of paper), and thus invariably selecting all examples with higher perplexity would include the noisier examples in the final labeled pool as well.
>
>
> - Perplexity Reweighting:
>    - The perplexity reweighting method is not intended to directly select examples with higher quality scores. Instead, its purpose is to systematically bring forward underrepresented instances for selection in subsequent iterations and increase the diversity. The selection of quality examples is done through the second stage of LLM quality filtering, rather than through the PPL reweighting.
>
>
> > Unclear why the perplexity - LLM pipeline is needed: Sachdeva et al., 2024 stated that the usage of LLM instead of perplexity to select quality data is because LLM is superior. The author has not justify their decision to use both methods in contrary to Sachdeva et al., 2024
>
> Calculating scores for the complete unlabeled dataset through LLM is significantly more computationally expensive due to the LLM inference costs, as mentioned in lines 75-77 of our paper, and also highlighted by Sachdeva et al. in Section 5 of [1]. Therefore, we design a two-stage pruning process in ActivePrune, through which the perplexity calculation via n-gram LM (KenLM) is done for the complete unlabeled pool (which is significantly faster), and only a **small portion of the unlabeled pool** consisting of examples with high perplexity scores is sent to the quantized LLM for quality scores.
>
>
> > Figure 4 seems to suggests the Ask-LLM would be a better candidate as the chosen samples have higher quality and the mass of the selection examples are more spread out on the perplexity tail end compare to ActivePrune.
>
> We agree that while ASK-LLM picks useful and informative examples (as shown in Figure 4), this selection comes at the cost of a much higher computational cost, which makes it infeasible to be used directly for data pruning in AL. The results in Table 1 demonstrate that ActivePrune shows better performance compared to ASK-LLM across all datasets, in addition to being significantly more efficient.
>
> > The speed up baseline is unreasonable: ActivePrune uses both perplexity and LLM but the speed up is 100x compare to methods that only use LLM or perplexity.
>
> The speedup with ActivePrune is primarily due to the following factors:
> - We use an n-gram language model for perplexity calculation, which shows a **significantly lower computational cost** (1818 perplexity queries per millisecond). This efficiency is not possible with a regular LLM.
> - We use a two-stage pruning process, where the perplexity is computed for the complete unlabeled pool, but the LLM data quality scores are computed for a **small percentage of examples** with higher perplexity.
> - The perplexity reweighting procedure is a **lightweight procedure** that ensures that underrepresented instances are brought forward for selection without significant overhead.
>
> The actual wall-clock time comparison of the end-to-end active learning procedure shown in Table 2 demonstrates that a 100x speedup is indeed achievable with the design choices added above.
>
> References:
>
> [1] https://arxiv.org/abs/2402.09668

---

> ### Author Response · Authors · 2024-11-20
> **Response to Reviewer WdAB (Part 2)**
>
> > Q1: Are the selected high perplexity examples actually good for training?
>
> There are two scenarios where high perplexity examples are used for training by ActivePrune:
> - The examples with high perplexity are scored through LLM data quality scores and are only passed to the AL procedure if their quality score is high. Thus, such scores are informative and important for learning. A few such examples are shown in Tables 6 and 7 in the appendix.
> - The underrepresented instances that have high perplexity (and are significantly different from the labeled instances) are systematically brought forward for selection through the perplexity reweighting procedure. These instances are crucial for increasing the diversity of the batch selected for labeling.
>
> > Q2: Does the competitiveness for speed go away when the baselines also use quantized LLM?
>
> While quantization does play a role in the efficiency of ActivePrune, the actual competitiveness in speed for ActivePrune comes due to the two-stage pruning process, where the initial evaluation of the complete set is done through perplexity through an n-gram LM and a small percentage of examples are evaluated through LLM data quality scores. Using a quantized LLM for the baselines does not bring the computational cost close to ActivePrune, since the scores through baselines are still calculated on the complete dataset instead of a significantly smaller subset.
>
> Thank you once again for your constructive feedback. We believe the clarifications provided above address your concerns.

---

> > ### Comment · Reviewer_WdAB · 2024-11-25
> >
> > Thank you for clarifying my questions and comments. Although I'm still not convinced on the method's speed up compare to perplexity, the method does improve upon Ask-LLM. It also shows that there are merits to using perplexity as a diversity probe. Therefore, I have raised my score.

---

> > > ### Author Response · Authors · 2024-11-26
> > >
> > > Thank you for your response! We’re glad to hear that the clarifications were useful.
> > >
> > > In the revised version, we have now provided a more detailed explanation of ActivePrune’s speedup compared to the perplexity and ASK-LLM baselines (highlighted in blue on lines 435–439).

---

### Official Review · Reviewer_5xZ3 · 2024-11-08

**Soundness:** 3
**Presentation:** 2
**Contribution:** 2
**Rating:** 6
**Confidence:** 4

**Summary:**

The main contribution of the paper is the introduction of ActivePrune, a novel data pruning strategy that enhances the efficiency of active learning. ActivePrune uses a two-stage pruning process: (1) a fast evaluation using perplexity scores from an n-gram language model, followed by (2) a high-quality selection using data quality scores computed with a quantized large language model (LLM). Additionally, ActivePrune employs a novel perplexity reweighting technique to improve diversity by prioritizing underrepresented instances in the dataset.

**Strengths:**

1.  The proposed ActivePrune framework is a novel, efficient two-stage pruning process that claims a significant reduction in computational costs in active learning without compromising data selection quality.

2.  The paper introduces a unique perplexity reweighting technique that systematically prioritizes underrepresented instances, promoting diversity in the selection pool across iterations.

3. The re-weighting algorithm based on the already selected samples in previous iterations in very unique to ActivePrune and is a very good way to enforce diversity.

**Weaknesses:**

1. The authors should clearly include the definitions of $\mathcal{L}$, $\mathcal{F}$, L etc. which seems to be used interchangeably in the document. eg. In line 226, is L here same as $\mathcal{L}$ introduced in line 161 ?

2. ActivePrune uses 2 LLMs in its pipeline but continues to enjoy 97% higher efficiency over 'Random' sampling techniques. This is misleading. Can this be a function of large compute resources used in the paper ?

3. Authors claim the use of a 'high-quality' LLM to score examples. LLMs have been shown in recent literature to demonstrate hallucinations and spurious correlations which have a high probability to impact the quality of predictions. The authors should provide additional analysis on this to establish the impact of the paper. This impact is even more necessary since the paper claims to have used a quantized language model (line 213) in their pipeline.

4. In lines 184 the authors mention the selection bottom $k$ instances. Since the list is sorted in ascending order, the top-k examples should have the lowest perplexity. This is also observed in line 217 where the authors mention the selection of bottom k indices, which for a list sorted in descending order should be the top-k elements.

5. The gain in performance for datasets other than IT Domain seem to be very marginal. In some cases such as the classification task in AG News dataset, the change in F1 score over random is very incremental which weakens the contribution of ActivePrune.

**Questions:**

1. In line 219 should the expression for $\mathcal{F}$ be $\mathcal{F} \cup \mathcal{Q}_s$ ?

2. In Section 3 the authors call out the key components of ActivePrune. These seem to also be the main contributions of the paper and should be clearly called out earlier in the paper, preferably in Section 1.

3. In line 182, the line ends without a period '.' and should be included.

4. The authors perform only 5 rounds of AL for each task. Is this a common practice or has been decided based on some factors ? This would highly increase readability.

---

> ### Author Response · Authors · 2024-11-20
> **Response to Reviewer 5xZ3 (Part 1)**
>
> Thank you for your insightful comments! We appreciate the positive feedback regarding our work. Addressing the specific comments below:
>
> > W1: The authors should clearly include the definitions of L, F, L etc. which seems to be used interchangeably in the document.eg. In line 226, is L here same as L introduced in line 161?
>
> Thank you for the feedback. The L (labeled set) on line 226 is the same as L added in line 161. As suggested, we’ve improved the definitions for clarity.
>
> > W2: ActivePrune uses 2 LLMs in its pipeline but continues to enjoy 97% higher efficiency over 'Random' sampling techniques. This is misleading. Can this be a function of large compute resources used in the paper ?
>
> Thank you for your feedback. The 97% higher efficiency claim compares ActivePrune to other **score-based pruning methods** (as mentioned at the beginning of line 89), not to random pruning. The rationale for comparing the computational efficiency to other score-based pruning methods is that they demonstrate significantly better performance than random pruning in most settings, and ActivePrune is able to perform even better than them at a fraction of the computational cost. We have revised the text to make this distinction clearer.
>
> > W3: Authors claim the use of a 'high-quality' LLM to score examples. LLMs have been shown in recent literature to demonstrate hallucinations and spurious correlations which have a high probability to impact the quality of predictions..........
>
> Thank you for the thoughtful comment. To understand the extent of LLM hallucinations, we refer to the Hughes Hallucination Evaluation Model (HHEM), an open-source model designed to quantify hallucination rates in language models (with summarization as the target task). According to the public leaderboard [1], the hallucination rate for strong open-source models, including variants of Gemma-2B, is below 8% and the latest smaller open-source models go below 5%, e.g., Phi-3.5. This provides a reasonable bound on the likelihood of hallucination affecting data quality scores.
>
> Given this upper bound, we now discuss how ActivePrune would mitigate hallucinations, whenever they occur, for the small percentage of examples.
> - Before any LLM scoring, all examples undergo perplexity scoring using a lightweight n-gram LM. This initial filter ensures that only a small subset of examples proceed to the LLM scoring stage, further reducing the proportion of examples that can be affected by hallucination of LLMs.
> - After the LLM scoring, the perplexity reweighting procedure ensures that examples that are repeatedly overlooked by the selection pipeline – including those that are potentially affected by hallucinated scores – are systematically brought forward for selection in future iterations. This approach reduces the likelihood that hallucinations will disproportionately affect the selection process.
>
> We believe this two-tiered scoring system **effectively minimizes the risk of hallucinations impacting the quality of selected examples**. Quantifying the hallucination rates for the LLM quality scoring procedure using ground-truth scoring datasets is an interesting dimension to explore further, and we will include this in future work.
>
> [1] https://github.com/vectara/hallucination-leaderboard
>
> > W4: In lines 184 the authors mention the selection bottom k instances. Since the list is sorted in ascending order, the top-k examples should have the lowest perplexity. This is also observed in line 217 where the authors mention selection of bottom k indices, which for a list sorted in descending order should be the top-k elements.
>
> Thank you for the feedback. We would like to clarify the bottom-k terminology that we have used in the paper.
>
> - Bottom-k on line 184 refers to the selection of the indices *1…k*, which correspond to the examples with the lowest perplexity when the list is sorted in ascending order.
> - Bottom-k on line 217 also refers to the selection of the indices *1…k*, which correspond to the examples with the highest data quality scores when the list is sorted in descending order.
>
> We understand that the top/bottom wording may be confusing, and have now made it clearer with better terminology using indices only (*1…k*).
>
> > W5: The gain in performance for datasets other than IT Domain seem to be very marginal. In some cases such as the classification task in AG News dataset, the change in F1 score over random is very incremental which weakens the contribution of ActivePrune.
>
> We would like to emphasize that the performance differences between ActivePrune and the baselines are **statistically significant** (line 400-402) across datasets, including AESLC, IMDB, and AG News, as shown over multiple runs. Furthermore, ActivePrune offers substantial computational efficiency, achieving a **superior accuracy-efficiency trade-off** compared to existing pruning strategies. This balance between accuracy and efficiency is a key contribution of our approach.

---

> > ### Author Response · Authors · 2024-11-20
> > **Response to Reviewer 5xZ3 (Part 2)**
> >
> > > Q1: In line 219 should the expression for F be F∪Qs ?\
> >
> > Thanks for pointing this out. This is fixed now in the paper.
> >
> > > Q2: In Section 3 the authors call out the key components of ActivePrune. These seem to also be the main contributions of the paper and should be clearly called out earlier in the paper, preferably in Section 1.
> >
> > Thank you for the feedback. Lines 79-84 in Section 1 mention these components. We have outlined them more clearly in the revised version now.
> >
> > > Q3: In line 182, the line ends without a period '.' and should be included.
> >
> > Updated.
> >
> > > Q4: The authors perform only 5 rounds of AL for each task. Is this a common practice or has been decided based on some factors? This would highly increase readability.
> >
> > The decision to use five rounds of Active Learning (AL) follows the protocol established in prior works on AL that are mentioned in the paper. We have clarified this in the revised version to improve readability and provide additional context.
> >
> > We hope these clarifications address the reviewer's concerns.

---

> > > ### Comment · Reviewer_5xZ3 · 2024-11-25
> > > **Thanks to Authors and Response to Rebuttals**
> > >
> > > I would like to thank the authors for the rebuttal and additional insights which have clarified a significant portion of my concerns/ questions. Some concerns regarding the impact of hallucinations and spurious correlations in quantized LLMs (left to future research by authors) and the large compute efficiency claimed in the paper, even with the inclusion of two LLMs (although claimed to be better than other score based techniques) continue to remain.
> > >
> > > Nevertheless, I will agree that adopting perplexity as a method for model pruning is a novel direction and have adjusted my ratings accordingly.
> > >
> > > On a side note, I would like to request the authors to highlight the changes made to manuscript during the rebuttal in a different color (or any other means) so that it is clearly distinguishable from the previously reviewed content.

---

> > > > ### Author Response · Authors · 2024-11-25
> > > >
> > > > Thank you for your response! We’re pleased to hear that the additional insights addressed most of your concerns. We'll further elaborate on the impact of hallucinations and the computational efficiency in the revised version as well.
> > > >
> > > > As suggested, we'll also highlight the changes made to the manuscript during the rebuttal in a different color.

---

### Author Response · Authors · 2024-11-24
**A gentle reminder for discussion on the rebuttal**

We sincerely thank the reviewers for their insightful comments and valuable feedback. We hope our responses have adequately addressed the concerns raised. If there are any additional comments or questions, we would be happy to address them during the discussion period.

Thank you!

---

### Author Response · Authors · 2024-12-03

As the discussion period concludes, we sincerely thank all the reviewers for their active engagement in the rebuttal discussion and revising the scores. We are glad that the clarifications and additional experiments addressed the raised concerns.

The revised version incorporates the following changes based on the suggestions of the reviewers. These changes are highlighted in blue colour in the uploaded version.

1. ⁠Added an explanation of ActivePrune’s speedup and computational efficiency compared to the perplexity and ASK-LLM baselines (lines 435–439).
2. ⁠Incorporated additional references in the related work section (Section 2.1 and 2.2).
3. ⁠Conducted experiments on an additional dataset from the medical domain in Appendix C.1.
4. ⁠Included SummaC scores for the summarization task and COMET scores for the translation task in Appendix C.1 and C.2.
5. Computed and presented the diversity score in Appendix C.3 to empirically demonstrate the inclusion of underrepresented instances.
6. ⁠Evaluated the combination of ActivePrune with the Contrastive Active Learning (CAL) strategy in Appendix C.4.
7. ⁠Evaluated ActivePrune with an additional model, BERT, in Appendix C.5.
8. Added a comparison of ActivePrune with k-means clustering as a data pruning method in Appendix C.6.
9. Provided further clarification on the robustness of perplexity and the LLM scoring procedure across various domains in Sections 3.2 and 3.3.

---

### Meta-Review · Area_Chair_62eP · 2024-12-20

**Metareview:**

This work introduces ActivePrune, a plug-and-play strategy for pruning large unlabeled data pools in active learning. Using a two-stage process—initial pruning with n-gram perplexity scores and fine-grained selection with quantized LLM metrics—it reduces computational costs while maintaining high-quality instance selection. A novel perplexity reweighting method further enhances diversity, and experiments across multiple tasks show that ActivePrune outperforms existing pruning methods.

The reviewers have raised a number of issues including the evaluation and baseline comparisons, and a few other issues. Though these have been addressed to some extent in the rebuttal, I would encourage the authors to address all the reviewer's criticisms carefully in the next version and resubmit.

**Additional Comments On Reviewer Discussion:**

The reviewers have raised a number of issues including the evaluation and baseline comparisons, and a few other issues. Though these have been addressed to some extent in the rebuttal, I would encourage the authors to address all the reviewer's criticisms carefully in the next version and resubmit.

---

### Decision · Program_Chairs · 2025-01-22

Reject